## Registered report

psychology/neuroscience/cognition

visual short-term memory, working memory, attention, N-back, EEG, inverted encoding model

**Author for correspondence:**
Quan Wan
e-mail: qwan22@wisc.edu

# Tracking stimulus representation across a 2-back visual working memory task

Quan Wan[1], Ying Cai[3], Jason Samaha[4] and Bradley R. Postle[1,2]

[1]Department of Psychology, and [2]Department of Psychiatry, University of Wisconsin–Madison, Madison, WI, USA
[3]Department of Psychology and Behavioral Science, Zhejiang University, Hangzhou, Zhejiang, People's Republic of China
[4]Department of Psychology, University of California, Santa Cruz, CA, USA

QW, 0000-0001-5743-086X; YC, 0000-0001-5125-2556; JS, 0000-0001-8010-5993; BRP, 0000-0001-8555-0148

How does the neural representation of visual working memory content vary with behavioural priority? To address this, we recorded electroencephalography (EEG) while subjects performed a continuous-performance 2-back working memory task with oriented-grating stimuli. We tracked the transition of the neural representation of an item ($n$) from its initial encoding, to the status of 'unprioritized memory item' (UMI), and back to 'prioritized memory item', with multivariate inverted encoding modelling. Results showed that the representational format was remapped from its initially encoded format into a distinctive 'opposite' representational format when it became a UMI and then mapped back into its initial format when subsequently prioritized in anticipation of its comparison with item $n + 2$. Thus, contrary to the default assumption that the activity representing an item in working memory might simply get weaker when it is deprioritized, it may be that a process of *priority-based remapping* helps to protect remembered information when it is not in the focus of attention.

## 1. Introduction

One critical property of cognitive control is its ability to prioritize stored memory content based on its relevance for the immediate behavioural context. However, how the neural representation of information held in working memory changes with its priority status remains poorly understood. Research exploring the effect of prioritization of working memory often employs a retrocueing task. A dual serial retrocueing (DSR) task begins with the presentation of two sample items, followed by a retrocue that

signals which of the two items will be tested first, rendering it the 'prioritized memory item' (PMI).[1] Because there is a 0.5 probability that the initially unprioritized memory item (UMI) may be tested later in the trial, it is assumed that the UMI is retained in working memory. A number of previous studies using multivariate pattern analysis (MVPA) of functional magnetic resonance imaging (fMRI) and electroencephalography (EEG) data have shown that classifier evidence for an active representation of the UMI drops to baseline levels (with fMRI [1–3] and with EEG [3,4]), findings that are at odds with the idea that all items held in working memory are represented with an active trace. The fact that this initially uncued item remains in working memory is demonstrated in several ways. First, an active trace of this item returns when it is cued by the second cue of the DSR task, and recognition performance on such 'cue-switch' trials is almost as high as is performance on the second probe of 'cue-stay' trials, when the same item is cued twice (e.g. [1,2,4]). Second, when a single pulse of TMS is applied during the first delay, MVPA evidence of the UMI is transiently elevated, and on these trials, the false-alarm rate for UMI foil probes is higher than it is on no-TMS trials [3]. It has been proposed that the UMI could be represented via 'activity-silent' mechanisms (e.g. [5]), perhaps, for example, through short-lived synaptic modifications [6].

More recently, however, studies presenting evidence for an active trace of the UMI have begun to emerge. In one fMRI study, Christophel *et al*. [7] reported that although an active trace of only the PMI can be decoded from early visual areas V1–V4, the UMI (together with the PMI) can be decoded from the intraparietal sulcus and frontal eye fields. From this, they suggested that 'sensory cortex maintains a high-resolution representation of the currently attended memory item, whereas parietal cortex has low-resolution representations of both attended and unattended items' (p. 496). A different possibility, one that we will refer to as 'priority-based recoding' is suggested by two other recent fMRI studies. In one study, van Loon *et al*. [8] presented two search targets, each one relevant for one of two subsequent and serially presented search arrays. They found that, during the first search array, the target that would be relevant for the second array (i.e. the item that was momentarily the UMI) could be decoded from the posterior fusiform cortex, but with a voxel pattern that was anticorrelated with its voxel pattern when it was relevant for a search (i.e. when it was the PMI). A second study, from our group, has produced a qualitatively similar pattern of results: using DSR for oriented gratings and analysing the resultant fMRI data with multivariate inverted encoding models (IEMs), Yu, Teng & Postle [9] have found that an active representation of the UMI can be reconstructed from signal from early visual cortex, but that this reconstruction is 'opposite' to its reconstruction when it is a PMI. More specifically, these 'opposite' IEM reconstructions can be equally validly characterized as being either (i) negative relative to the trained IEM, or (ii) shifted by 90 degrees relative to the trained IEM. Thus, the results of both van Loon *et al*. [8] and Yu, Teng & Postle. [9] suggest that the brain may use a recoding strategy to retain unprioritized information in working memory. Furthermore, this possibility is supported by the recent work on macaque neurophysiology, suggesting that neurons in the prefrontal cortex might 'morph population codes' in the face of a distractor to preserve information in a spatial working memory task [10,11].

In the present experiment, we propose to investigate whether the phenomenon of priority-based recoding of the UMI will extend to a novel task—a continuous-performance 2-back working memory task—with a neurophysiological technique that has not yet been used to address this question—EEG. Our motivations are manifold. *First*, because both dual serial visual search [8] and DSR working memory [9] tasks are administered in multiple stand-alone trials, the continuous-performance 2-back task, with its requirement to sustain task engagement across tens of seconds and to update the contents of working memory with each new stimulus, is an effective task with which we can assess the generality of the hypothesized phenomenon of priority-based recoding. *Second*, due to its superior temporal resolution, EEG can provide novel insight into the temporal dynamics of the putative recoding processes. *Third*, and of relevance to Yu, Teng & Postle [9] and to many other studies that have used line orientation as stimuli in tests of visual working memory, it has recently been shown that subtle but systematic differences in location of gaze can be used to decode working memory for line orientation and may therefore represent a confound for studies relying on MVPA of line orientation [12]. Alternatively, rather than framing such results as an empirical confound, it has been proposed that microsaccades (and, indeed, microsaccade-related activity in early visual cortical areas) play a functional role in encoding the contents of non-spatial visual working memory [13]. In view of these developments (and although it seems unlikely to us that an eye position strategy could be successfully

---

[1]Note that, although our group has previously used the terminology 'attended memory item (AMI)' and 'unattended memory item (UMI)', here we replace 'attended' with 'prioritized', because the latter refers to the experimental manipulation, without assuming a cognitive state.

implemented during a 2-back task when two different items are always being held simultaneously), we planned to simultaneously collect infrared (IR)-based eye tracking data and electrooculogram (EOG) data to investigate possible contributions of microsaccades to non-spatial 2-back working memory performance. *Fourth*, and finally, the results from a preliminary EEG study ($N = 12$) that served as a starting point for this registered report highlight a quality of the 2-back task that makes it well suited for the study of priority-based recoding: it affords the tracking of the systematic trajectory that a mnemonic representation takes through different states of priority.

When an item $n$ is presented within a block of 2-back stimuli, it first acts as a probe against which the memory of item $n - 2$ must be compared. After the $n - 2$-to-$n$ decision, $n$ is held in working memory as a UMI, while item $n - 1$ is prioritized for its impending comparison against item $n + 1$. Next, after the $n - 1$-to-$n + 1$ comparison, item $n$ is prioritized for its impending comparison against item $n + 2$. (The question of whether, after the $n$-to-$n + 2$ comparison, item $n$ is then actively removed from working memory, or whether its representation passively decays, is currently poorly understood, but is outside the scope of the present registered report.) To track the neural correlates of this trajectory, in a pilot study, we first trained an IEM to reconstruct stimulus orientation from EEG recorded during the stimulus presentation epoch and the delay epoch of an independent one-item delayed-recognition task. This first step revealed that a stable neural representation of the stimulus-evoked 'perceptual' representation was sustained across most of the delay period (electronic supplementary material, figure S1). Next, when an IEM trained on data from the early-delay period in the delayed-recognition task (i.e. a 'perceptual' IEM) was tested on EEG data from the 2-back task, we observed an evolution of the IEM reconstruction of item $n$ that was generally consistent with the expected operational trajectory of the 2-back task: first, very soon after the $n - 2$-to-$n$ button press, IEM reconstruction of item $n$ was opposite to its perceptual representation; subsequently, very soon after the $n - 1$-to-$n + 1$ button press, the IEM reconstruction of item $n$ transformed back into its perceptual representational format (electronic supplementary material, figure S2). We interpret this sequence as a reflection of priority-based recoding: first, once the $n - 2$-to-$n$ decision was made, the mnemonic representation of item $n$ was deprioritized (and so its neural representation was recoded into a format opposite of its perceptual representation); subsequently, once the $n - 1$-to-$n + 1$ decision was made, the mnemonic representation of item $n$ was reprioritized (and so its neural representation was decoded back into its perceptual representational format). If this account of priority-based recoding were to be confirmed in the registered report proposed here, these results would have important implications for our understanding of the neural and cognitive bases of prioritization in working memory.

In this article, we will make a distinction between the 'Pilot Study' and the 'Registered Report': '*Pilot Study*' will refer to the preliminary study ($N = 12$) summarized earlier, the results from which we generated hypotheses and prospectively estimated the power needed to test the hypotheses that are the focus of this '*Registered Report*'. There were four reasons that we decided to use the data from the *Pilot Study* as a hypothesis-generating dataset for this *Registered Report*. First, the results from the *Pilot Study* provided the bases for explicit, *a priori* predictions about the effects that priority-based recoding will have on IEM reconstructions of stimulus representations, as well as the data from which to estimate prospectively the number of subjects needed to achieve sufficient power to observe these predicted results. Second, a flaw in the stimulus selection procedure in the *Pilot Study*—requiring a minimum angular distance between adjacent items—made the interpretability of negative/shifted IEM reconstructions equivocal, and the design of the *Registered Report* avoided this flaw. Third, high-resolution IR-based eye tracking data were not collected during the *Pilot Study*, making it impossible to assess the relation of microsaccades to task performance and to multivariate analyses of the EEG data. Finally, without a control condition, which was lacking in the *Pilot Study*, the 2-back results were amenable to the theoretically trivial alternative explanation that the negative/shifted IEM reconstructions observed while an item is a UMI might merely have been a result of a post-stimulus undershoot in the EEG signal. This was addressed in the *Registered Report* by including a 1-back condition as a control. (The 1-back includes the continuous-performance properties that differentiate N-back tasks from stand-alone-trial tasks, but unlike the 2-back, the 1-back does not include an epoch when item $n$ takes on the status of UMI.)

## 1.1. Preregistered hypotheses

We proposed to test three *principal hypotheses* and eleven[2] *secondary hypotheses* in this *Registered Report*. The principal hypotheses addressed the proposed core mechanism of priority-based recoding

---

[2]Note that in the accepted Stage 1 Registered Report this was mistakenly listed as nine.

(including the 1-back control task). The secondary hypotheses related to the role of microsaccades and/or location of gaze in representing visual working memory for orientation.

### 1.1.1. Principal hypotheses

#### 1.1.1.1. Principal hypothesis 1 (2-back)
The IEM reconstruction of an item $n$'s orientation from the EEG signal from the 2-back task, averaged across the span from 1150 to 3150 ms after $n$'s onset (i.e. during the portion of the interstimulus interval (ISI) between item $n$ and item $n + 1$ when item $n$ is presumed to be a UMI), will be negatively correlated with the basis function used to train the IEM on data from 940 to 1040 ms after stimulus onset of the 1-item delayed-recognition task (i.e. when a UMI, the representational format of item $n$ will be the opposite of its 'perceptual representation'). The precise method we will use to access the correlation with the basis function is described in §2.3.1.2.

#### 1.1.1.2. Principal hypothesis 2 (2-back)
The IEM reconstruction of item $n$'s orientation from the EEG signal from the 2-back task, averaged across the span from 1150 to 3150 ms after item $n + 1$'s onset (i.e. during the portion of the ISI between item $n + 1$ and item $n + 2$ when item $n$ is presumed to be a PMI), will be positively correlated with the basis function used to train the IEM on data from 940 to 1040 ms after stimulus onset of the 1-item delayed-recognition task (i.e. when a PMI, the representational format of item $n$ will be the same as its 'perceptual representation').

#### 1.1.1.3. Principal hypothesis 3 (1-back)
The IEM reconstruction of an item $n$'s orientation from the EEG signal from the 1-back task, averaged across the span from 1150 to 3150 ms after $n$'s onset (i.e. during the portion of the ISI between item $n$ and item $n + 1$, when item $n$ is presumed to be a PMI), will be positively correlated with the basis function used to train the IEM on data from 940 to 1040 ms after stimulus onset of the 1-item delayed-recognition task (i.e. the representational format of an item held in visual working memory ('perceptual' versus 'opposite') is determined by its priority status, not by time elapsed since its offset, as would be predicted by an undershoot account of the prediction from *principal hypothesis 1*.)

### 1.1.2. Secondary hypotheses

#### 1.1.2.1. Secondary hypothesis 1 (1-back)
IEM training on 1-back data, using $k$-fold cross-validation, will produce results that are qualitatively similar to those produced when testing a delayed-recognition task–trained IEM on these data (i.e. similar to those generated by the test of *principal hypothesis 3*). (The rationale is that because the 1-back task only requires the retention in working memory of a single item, this approach is more likely to be successful than it would be if applied to the 2-back data. Unlike the procedure that will be used to test *principal hypothesis 3*, the $k$-fold cross-validation approach does not make the assumption that the neural code used during the 1-back task will be the same as that used during the delayed-recognition task).

#### 1.1.2.2. Secondary hypothesis 2 (2-back)
During stimulus presentation, location of gaze will vary systematically with the stimulus item's orientation, such that its identity can be decoded from eye position alone. The precise method we will use to decode with eye position is described in §2.3.2. (If this hypothesis is confirmed, we will then assess whether perception-related eye position can also be used to decode the identity of the information being held during the ISI (i.e. in visual working memory).)

#### 1.1.2.3. Secondary hypothesis 3 (2-back)
During the 2-back task, location of gaze during the period spanning from 1150 to 3150 ms after $n$'s onset will vary systematically with $n$'s orientation, such that the identity of $n$ can be decoded from eye position alone. (Confirmation of this hypothesis would suggest that the UMI is encoded in eye position.)

### 1.1.2.4. Secondary hypothesis 4 (2-back)

During the 2-back task, location of gaze during the period spanning from 1150 to 3150 ms after $n$'s onset will vary systematically with item n −1's orientation, such that the identity of $n − 1$ can be decoded from eye position alone. (Confirmation of this hypothesis would suggest that the PMI is encoded in eye position.)

### 1.1.2.5. Secondary hypothesis 5 (2-back)

During the 2-back task, location of gaze during the period spanning from 1150 to 3150 ms after $n$'s onset will alternate systematically between locations corresponding to the orientations of $n − 1$ and of $n$, such that the identity of both can be decoded from eye position alone. (Confirmation of this hypothesis would suggest that the multiple items can be encoded by eye position during a single retention period.)

### 1.1.2.6. Secondary hypothesis 6 (1-back)

During stimulus presentation, location of gaze will vary systematically with the stimulus item's orientation, such that its identity can be decoded from eye position alone. (This would be a replication of *secondary hypothesis 2*. If it is confirmed, we will then assess whether perception-related eye position can also be used to decode the identity of the information being held during the delay period (i.e. in visual working memory).)

### 1.1.2.7. Secondary hypothesis 7 (1-back)

During the 1-back task, location of gaze during the period spanning from 1150 to 3150 ms after $n$'s onset will vary systematically with $n$'s orientation, such that the identity of $n$ can be decoded from eye position alone. (Confirmation of this hypothesis would be consistent with the idea that items in non-spatial visual working memory are encoded, at least in part, in eye position.)

### 1.1.2.8. Secondary hypothesis 8 (delayed recognition)

During stimulus presentation, location of gaze will vary systematically with the stimulus item's orientation, such that its identity can be decoded from eye position alone. (This would be a replication of *secondary hypotheses 2* and *6*. If it is confirmed, we will then assess whether perception-related eye position can also be used to decode the identity of the information being held during the delay period (i.e. in visual working memory).)

### 1.1.2.9. Secondary hypothesis 9 (delayed recognition)

During the delayed-recognition task, location of gaze during the delay period spanning from 1000 to 2000 ms[3] after $n$'s onset will vary systematically with $n$'s orientation, such that the identity of $n$ can be decoded from eye position alone. (Confirmation of this hypothesis would be consistent with the idea that items in non-spatial visual working memory are encoded, at least in part, in eye position.)

### 1.1.2.10. Secondary hypothesis 10 (2-back)

The IEM reconstruction of item $n$'s orientation from the EEG signal from the 2-back task during the presentation of the non-match item $n + 1$ will be positively correlated with the basis function used to train the IEM on data from 940 to 1040 ms after stimulus onset of the 1-item delayed-recognition task. (Confirmation of this hypothesis would be consistent with the idea that visual stimulation alone, even unrelated, can recode a UMI back into its 'perceptual representational format'.)

### 1.1.2.11. Secondary hypothesis 11 (1-back)

The IEM reconstruction of item $n$'s orientation from the EEG signal from the 1-back task averaged across the span from 1150 to 3150 ms after the non-match item $n + 1$'s onset (i.e. during the portion of the ISI between item $n + 1$ and item $n + 2$, when item $n$ is no longer relevant to the task) will be uncorrelated with the basis function used to train the IEM on data from 940 to 1040 ms after stimulus onset of the 1-item delayed-recognition task. (Observing a negative reconstruction of item $n$ will serve as evidence against our main hypothesis that the negative reconstruction is a consequence of deprioritization

---

[3]Note that in the accepted Stage 1 Registered Report this was mistakenly listed as 1150–3150 ms.

among competing task-relevant representations rather than merely of an item ($n$) being superseded in priority by another item ($n + 1$).)

# 2. Methods

## 2.1. Subjects

Thirty healthy young adults (19 females; mean age = 21.6 years, s.d. = 4.2 years) from the University of Wisconsin–Madison community participated in the EEG experiment. Using the data from the *Pilot Study*, as well as data from two previous studies from our group, to estimate effect sizes indicated that we needed data from 30 subjects to achieve 90% power to detect the effect as predicted by both *principal hypothesis 1* and *principal hypothesis 2*.

Subjects met the inclusion criteria that included an age range of 18–35 years, being right-handed, having normal or corrected-to-normal vision, reporting no history of neurological disease, seizures or fainting, nor use of psychotropic drugs, nor chronic consumption of alcohol. In addition, subjects were only selected for participation in the EEG experiment if, for at least one of six 38-item training blocks administered during behavioural training, they achieved an accuracy of 80% correct or higher. Subjects were excluded from analyses if (i) their performance accuracy in any task was more than 3 standard deviations below the group average, or (ii) less than 50 trials were left in any orientation condition after EEG data cleaning. All subjects provided written consent approved by the University of Wisconsin–Madison Health Sciences Institutional Review Board. All subjects were monetarily compensated for their participation for the behavioural training and the EEG sessions.

## 2.2. Stimuli and procedure

### 2.2.1. Training/screening

Stimuli were black-and-white sinusoidal gratings (contrast = 0.3, spatial frequency = 1 cycle/°; phase angle varying randomly between 0° and 180° for each presentation) presented centrally within a circular patch (radius = 2.8°).[4] The six stimulus orientations were 10°, 40°, 70°, 100°, 130° and 160° (cardinal orientations avoided to reduce the likelihood of verbal encoding). Masks were radial chequerboards of the same size and positioning as the stimuli (figure 1*a*). All stimuli were generated and presented in Matlab (MathWorks) and Psychtoolbox-3 extensions (http://psychtoolbox.org [14]).

The training/screening session, which took place on a separate day prior to the EEG session, comprised six 38-stimulus blocks of the 1-back task and six 38-stimulus blocks of the 2-back task in a randomly determined order. Subjects were encouraged to encode the stimulus orientations visually rather than to employ verbal or gestural strategies. All received payment upon completion of the session and those who achieved an accuracy of 80% or above in at least one of the six blocks of 2-back were contacted on a subsequent day and invited to participate in the EEG session.

### 2.2.2. Electroencephalography session

#### 2.2.2.1. Behavioural tasks

The EEG session consisted of four blocks of the 2-back task randomly interleaved with four blocks of a 1-back task and six blocks of the functional localizer task, followed by eight blocks of 1-item delayed recognition.

*2-back task*. Each block of the 2-back task contained 128 serially presented stimuli. Each item was presented for 500 ms, followed by a 50 ms ISI, then a 200 ms mask and finally an ISI (or 'delay period') that varied unpredictably between 2.8 and 3.2 s (duration randomly chosen from a vector spanning a range from 2800 to 3200 ms in 50 ms increments; figure 1*a*).

For each subject, the stimulus sequences for each block were generated via an algorithm that pseudorandomly selects stimuli from a list of 126 stimulus-match condition pairs (7 'match' and 14 'non-match' for each orientation), such that each item is followed by each of the six possible items equiprobably. In this way, the stimulus sequences experienced by each subject were unique. In addition, each stimulus sequence met the constraint that no stimulus could appear as a match more

---

[4]Note that in the accepted Stage 1 Registered Report this was mistakenly listed as 5°.

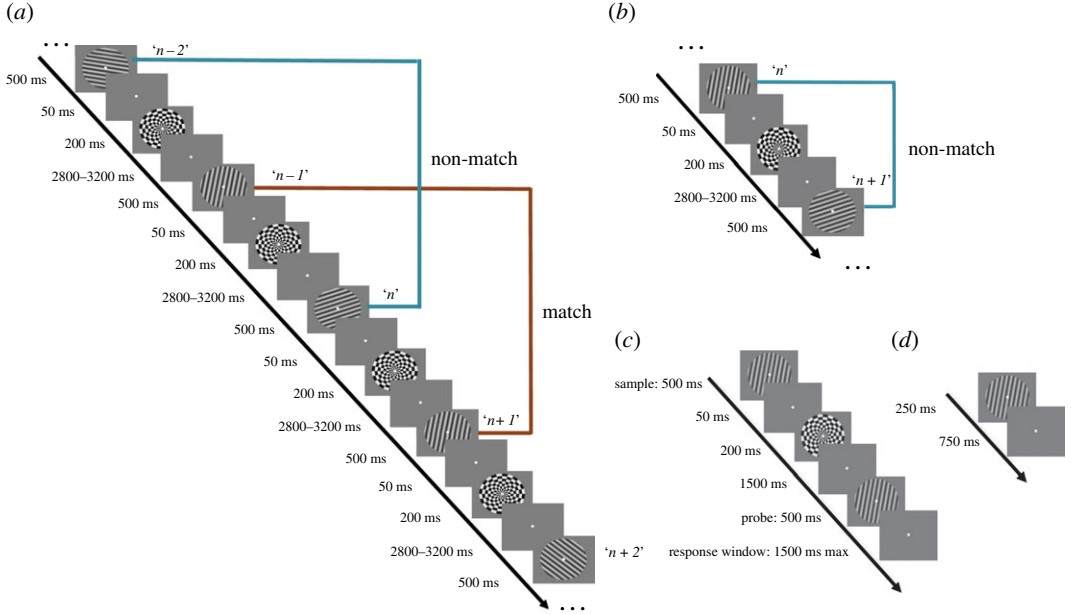

**Figure 1.** Experimental procedure. (*a*) 2-back task. (*b*) 1-back task. (*c*) Delayed-recognition task. (*d*) Functional localizer task.

than two consecutive times (e.g. a sequence like 'A B A C A D A' was not permitted). Critically, the constraint from the *Pilot Study* that the same orientation could not occur twice in a row (i.e. 'no 1-back matches') was not applied to the *Registered Report*. This is an important detail because IEM training can be artefactually biased if a minimum distance is imposed between consecutive stimuli.

Subjects were given brief practice before the beginning of each task. For the 2-back task, subjects were instructed to respond promptly to the onset of each stimulus, indicating via a key press whether the orientation of the most recently presented stimulus was the same as (match) or different from (non-match) the stimulus presented two positions previously. The mapping of 'F' and 'J' keys on the keyboard to 'match' and 'non-match' responses was counterbalanced across subjects.

*1-back task*. All details for the 1-back task were the same as for the 2-back task, with the exceptions that each block had 127 stimuli and that 'match' and 'non-match' labels were determined according to the 1-back matching status (figure 1*b*).

*Delayed-recognition task*. This task was administered to generate data on which to train IEMs that would then be tested on the 2-back data. In particular, the delay period of the 1-item delayed recognition was expected to generate better IEM models than would the ISIs from the 2-back task due to the additional memory load and cognitive demands of the latter. Each trial started with the central presentation, for 500 ms, of a sample item, followed by a 50 ms ISI and then a 200 ms presentation of a radial chequerboard mask (same sequence and stimulus specifications as the 2-back task; figure 1*c*). After a 1.5 s delay period, a probe was displayed for 500 ms, with subjects instructed to promptly press the 'F' or 'J' key to indicate 'match' or 'non-match' of the probe to the sample (same key-to-response mapping that the subject had used for the 2-back task). Matching probes were identical to that trial's sample, and non-matching probes were rotated 15° from the sample orientation (clockwise/counterclockwise rotations equiprobable). The response window was terminated either when a key press was registered, or when 1.5 s had elapsed, with non-responses scored as errors. The intertrial interval (ITI) began with 300 ms of blank screen, followed by a central fixation target for a variable 500–800 ms (duration randomly chosen from a vector spanning a range from 500 to 800 ms in 50 ms increments).

The eight blocks of delayed recognition each had a unique randomized sequence of 72 trials, with each of the six stimuli from the 2-back task appearing as the sample on 12 trials (six times paired with a matching probe, six times with a non-matching probe), with sample identity and 'match' and 'non-match' status both occurring unpredictably.

*Functional localizer task*. Following the methods of Mostert *et al*. [12], six functional localizer blocks were interleaved with the 1-back and 2-back tasks. Each block consisted of 120 trials, in which a grating with one of the six orientations was equiprobably presented for 250 ms, followed by a 750 ms inter-trial interval (figure 1*d*). Subjects were instructed to monitor the centre of the screen for brief flickers of the fixation dot occurring at random times during the block and to press a key whenever it happened. In each block,

the flicker occurred for a random number of times between 8 and 12. This task created a condition in which subjects maintained fixation and did not attend to the oriented-grating stimuli and was thus used to train and test IEMs assumed to be based on purely bottom-up, sensory signals in exploratory analyses.

### 2.2.2.2. Electroencephalography data acquisition and preprocessing

Concurrent with all the tasks, EEG data were recorded at 1000 Hz with a Brain Products (Munich, Germany) 64-channel actiCAP. Electrodes were referenced against an electrode placed at the centre of the forehead, and impedances were kept below 15 kΩ. Eye movements were monitored with EOG electrodes placed near the external canthus of, and below, the right eye.

EEG preprocessing was performed using EEGLAB [15]. Bad EEG channels were identified by visual inspection and then removed and interpolated using the 'spherical' method. EEG data were bandpass-filtered from 0.01 to 80 Hz and notch-filtered at 60 Hz. The 2-back data were cut into epochs beginning 200 ms prior to and ending 3550 ms after each stimulus item's onset (i.e. ending 2800 ms into the ISI, the longest duration for which there were data from all trials; the 200 ms prior to the item's onset was used to determine baseline). The 1-back data were cut into epochs beginning 200 ms prior to and ending 3550 ms after each item's onset. The delayed-recognition data were cut into epochs beginning 200 ms prior to and ending 2750 ms after each sample item's onset (i.e. until the offset of the probe). The functional localizer data were cut into epochs beginning 200 ms prior to and ending 1000 ms after each item's onset. All epochs were baseline corrected and re-referenced to the median. Epochs with eye-related or muscle-related artefacts were identified via visual inspection of the raw data and of components generated from an independent component analysis decomposition and were removed prior to IEM analyses.

### 2.2.2.3. Infrared-based eye tracking

Concurrent with EEG recording, gaze position and pupil size during the EEG session were recorded at 1000 Hz with an IR-based system (EyeLink 1000 with Long Range Mount; SR Research, Ottowa, ON).

To prepare for classification analyses, $X$ and $Y$ coordinates of gaze position data were first converted to visual angle and downsampled to 250 Hz. Outliers and artefacts were identified using a Hampel filter with a half filter window of 6 s, and within the filter window, any data that exceeded the threshold of 3.5 median absolute deviations above or below the median were replaced by the median of the window. In each block, any missing data were replaced by the median coordinates from that block, and then, the data were smoothed via a moving average with a 100 ms window. The first trial of each block was removed to correct for any systematic timing inaccuracy at the beginning of the block. Data were cut into epochs and baseline corrected in the same fashion as in the EEG data preprocessing prior to classification analyses.

## 2.3. Data analysis

### 2.3.1. Multivariate inverted encoding model

We used multivariate inverted encoding models [16–18] to evaluate the neural representation of stimulus orientation (figure 2). IEMs were trained and tested on voltages from all 60 electrodes. EEG data from all trials (both correct and incorrect) were included in the training and reconstruction of IEMs. The voltage recorded from each electrode could be construed as a weighted sum of responses from six hypothetical channels, each channel optimally tuned for a specific orientation (i.e. 10°, 40°, 70°, 100°, 130°, 160°). The tuning for each channel was modelled with a half-wave-rectified sinusoid raised to the sixth power:

$$R = \sin^6(x),$$

where $x$ is centred on the orientation to which the channel was optimally tuned and $R$ is the channel response. If we let $B_1$ ($m$ electrodes × $n$ trials) be the recorded voltages from each electrode in each trial from the training dataset, $C_1$ ($k$ channels × $n$ trials) the predicted responses from each tuning channel, and $W$ ($m$ electrodes × $k$ channels) the weight matrix that maps the data from 'channel space' to 'electrode space', the following equation describes the relationship between $B_1$, $C_1$ and $W$:

$$B_1 = WC_1.$$

We could estimate the weight matrix $\hat{W}$ using least-squares regression:

$$\hat{W} = B_1 C_1^T (C_1 C_1^T)^{-1}.$$

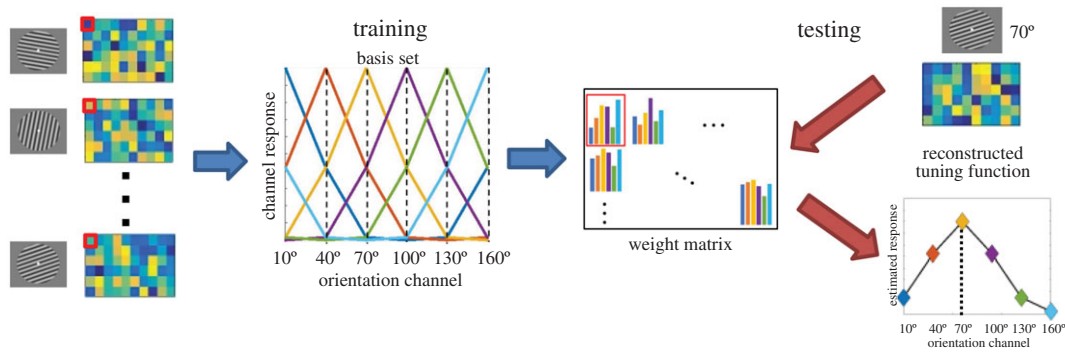

**Figure 2.** Multivariate IEM. Voltage from each EEG electrode can be construed as a weighted sum of responses from six channels (modelled by a half-wave-rectified sinusoid raised to the 6th power), each tuned to a specific stimulus orientation, comprising the basis set. By regressing the training data onto the basis set, we obtain a weight matrix that characterizes the contribution of each orientation channel to each electrode's response (i.e. we derive each electrode's tuning function). We then use the weight matrix to invert the model and test it on a novel dataset to derive the reconstructed representation of stimulus orientation.

By using the above weight matrix, we could invert the model, map the observed signal from an independent testing set $B_2$ ($m$ electrodes × $n$ trials) from the 'electrode space' to 'channel space' and reconstruct a set of estimated channel responses $\widehat{C_2}$ ($k$ channels * $n$ trials):

$$\widehat{C_2} = (\hat{W}^T\hat{W})^{-1}\hat{W}^T B_2.$$

Then, we could circularly shift the estimated channel responses to a common centre (0°) to derive the reconstructed tuning function.

### 2.3.1.1. Generating inverted encoding models from delayed-recognition task data

First, we downsampled the EEG voltages to 50 Hz and employed a $k$-fold cross-validation procedure (i.e. 'leave-one-run-out') to assess the robustness of the reconstruction of stimulus identity (i.e. orientation) across the delayed-recognition task. Each IEM was trained and tested on the same time point. In the *Pilot Study*, we observed strong stimulus reconstruction from time points spanning from soon after sample onset through the initial portion of the delay period. Notably, this pattern spanned the time during which the mask was presented, suggesting that the sample was robustly encoded into working memory; electronic supplementary material, figure S1). Next, we sought the optimal epoch from the delayed-recognition task on which to train the IEM that would be used to track stimulus representation across the 2-back task. We did this by training IEMs on EEG data averaged within a 100 ms window and by sliding this window in 20 ms increments across the entirety of the trial. (In our experience, a 100 ms window provides a good compromise between the countervailing desiderata of high temporal resolution versus robust IEM training; because this was explicitly carried out to determine optimal parameters for the *Registered Report*, correction for multiple comparisons was not applied.)

### 2.3.1.2. Inverted encoding model reconstruction of data from the 2-back task

Separately for each subject, we tested delayed-recognition-trained IEMs on the 2-back data to track the reconstruction of an item's tuning function across the two consecutive stimulus presentation events when it was in working memory. For each stimulus item $n$ (excluding the first and the last two items in the task block), this was done by 'stitching together' every two adjacent stimulus-processing epochs, beginning for each from 200 ms prior to the presentation of $n$ and ending 2.8 s after the mask presentation following item $n$ (see diagram of timeline of task events on left-hand side of figure 4). (Note that the two epochs were not continuous because only 2800 ms of each jittered ISI, following the offset of the mask, was shared in common by all stimulus epochs.) To improve signal-to-noise ratio, we smoothed channel output by means of a moving average with an 80 ms window. To evaluate the robustness of the reconstruction at each time point, we conducted a Pearson's correlation between channel responses of the reconstructed tuning function and the basis function separately for each subject and then tested the Fisher $z$-transformed correlation coefficients from all subjects against 0 using two-tailed one-sample $t$ tests with $\alpha = 0.05$. Although we did not make time-point-by-time-point predictions for this *Registered Report* (for which the principal hypothesis tests pertain to IEM reconstructions averaged across 2 s-long windows within ISIs), a cluster-based permutation test [19,20] was applied to data to provide a

descriptive feel for the temporal dynamics of the reconstructions. First, we identified clusters of contiguous time points that exceeded either a positive threshold of $t = 2.045$ or a negative threshold of $t = -2.045$ (corresponding to a $p$-value of 0.05, two tailed). This clustering procedure was done separately for time points with a positive or a negative $t$-value. A cluster-level test statistic was calculated for each cluster by summing all the $t$-values in the cluster. To perform the non-parametric statistical test on the group level, we constructed a null distribution by permuting for 10 000 iterations. In each iteration, after the correlation coefficients (between the reconstructed tuning curve and the basis function) were Fisher $z$-transformed, their signs were flipped randomly. Then, clusters were identified as per the aforementioned procedure. For each iteration, the largest cluster-level statistic (in absolute value) was used to construct the null distribution. Any clusters from the 2-back time course with a cluster-level statistic (in absolute value) larger than the 97.5th percentile of the null distribution was significant at an $\alpha = 0.05$, two tailed.

To evaluate stimulus representation of item $n$ during the ISIs following $n$ and following $n + 1$, we averaged the channel responses across a 2000 ms window centred on the two 2800 ms post-mask ISIs and ran the same $t$ test on Fisher $z$-transformed correlation coefficients. The $p$-values from this analysis were corrected across both windows using the false discovery rate (FDR) method. The results from these analyses in the *Pilot Study* served as a basis for generating hypotheses and calculating statistical power for this *Registered Report*.

### 2.3.1.3. Inverted encoding model reconstruction across 1-back task

We also reconstructed the stimulus orientation across the 1-back task with the same delayed-recognition task–trained IEM that was used for the 2-back data. In addition, we trained IEMs on the 1-back data using $k$-fold cross-validation.

### 2.3.1.4. Testing inverted encoding models trained on the functional localizer task data

IEMs built from the functional localizer task were also used to reconstruct the stimulus orientation in the delayed-recognition, 1-back and 2-back tasks as exploratory analyses supplementary to the procedures proposed for the primary and secondary hypothesis tests.

### 2.3.2. Infrared-based eye tracking data

To test the *secondary hypotheses* relating to eye position, we fed gaze position data to a multi-class probabilistic classifier to decode stimulus orientation. We followed the methods of Mostert *et al.* [12] due to its success in decoding stimulus orientation from the gaze position data. The six-class classifier was originally based on Bishop [21] (pp. 196–199, as cited in [12]). Specifically, we modelled the class-conditional densities as Gaussian distributions assuming equal covariance. Using Bayes' theorem while assuming a flat prior probability distribution, we inserted our gaze position data into the model and inverted it to generate posterior probabilities. If we let $x$ be a column vector with the length of the number of features, i.e. 2 (horizontal and vertical eye position coordinates), then we obtained the posterior probability that our observations belonged to class $k$ via the following equations:

$$P(\text{class} = k|\boldsymbol{x}) = \frac{\exp{(a_k)}}{\sum \exp{(a_j)}},$$

$$a_k(\boldsymbol{x}) = \boldsymbol{w}_k^T \boldsymbol{x} + w_{k0},$$

$$\boldsymbol{w}_k = \boldsymbol{S}^{-1} \boldsymbol{\mu}_k$$

and
$$w_{k0} = -\boldsymbol{\mu}_k^T \boldsymbol{S}^{-1} \boldsymbol{\mu}_k,$$

where $\boldsymbol{\mu}_k$ is the mean of class $k$ and $\boldsymbol{S}$ is the common covariance. Both variables were obtained from the training dataset. $\boldsymbol{S}$ was calculated as the unweighted mean of the six covariance matrices for each different class and then regularized using shrinkage ([22], as cited in [12]) with a regularization parameter of 0.01.

To evaluate the statistical significance of the decoding, we averaged the mean classification accuracy over the time windows specified in the *secondary hypotheses* and performed two-tailed one-sample $t$ tests on these average classification accuracy values for all subjects against chance (0.1667) with $\alpha = 0.05$. The $p$-values from the $t$ tests were FDR corrected for multiple comparisons across windows within each task. To provide a descriptive feel for the temporal dynamics of the decoding, we conducted similar $t$ tests time point by time point across the whole trial and corrected for multiple comparisons using the cluster-based permutation test method detailed in §2.3.1.2.

In the Stage 1 Registered Report, we mentioned that for instances of successful decoding of stimulus presentation data, we would test for generalization to delay period/ISI data. However, because no significant decoding of stimulus presentation data was obtained from any of the three working memory tasks, we did not perform any temporal generalization analyses.

# 3. Results

## 3.1. Behaviour

Subjects performed well on all four experimental tasks. For 2-back, accuracy was $85.7\% \pm 6.0\%$ (s.d.), $d'$ was $2.39 \pm 0.73$ and response time was $0.85 \pm 0.21$ s. For 1-back, accuracy was $92.3\% \pm 4.6\%$, $d'$ was $3.17 \pm 0.68$ and response time was $0.75 \pm 0.18$ s. For delayed recognition (DR), accuracy was $78.5\% \pm 7.0\%$, $d'$ was $1.86 \pm 0.59$ and response time was $0.66 \pm 0.12$ s. For functional localizer, accuracy was $99.8\% \pm 0.2\%$ and response time was $0.47 \pm 0.05$ s. All subjects satisfied the inclusion criteria for data analyses. One subject's behavioural data from two delayed-recognition task blocks were accidentally overwritten and hence not included in the analyses.

## 3.2. Testing hypotheses

### 3.2.1. Preprocessing

#### 3.2.1.1. Electroencephalography

Artefact rejection resulted in an average of 546 artefact-free trials per subject for the delayed-recognition task, 493 artefact-free post-stimulus ISIs for 2-back, 492 artefact-free ISIs for 1-back and 702 artefact-free trials for the functional localizer. An average of 91 trials per orientation per subject was analysed for the delayed-recognition task, 82 post-stimulus ISIs for 2-back, 82 ISIs for 1-back and 117 trials for the functional localizer task. (EEG data for the first two delayed-recognition blocks from one subject were discarded because of the overwritten behavioural data.)

#### 3.2.1.2. Gaze position

Some blocks of gaze position data were not included in the analyses: one 2-back block from a subject failed to save; data of one delayed-recognition block from another subject was incomplete; as mentioned earlier, behavioural and gaze position data of two delayed-recognition blocks of another subject were accidentally overwritten.

### 3.2.2. Delayed recognition

The $k$-fold cross-validation yielded a robust IEM reconstruction during the early-delay time window (940–1040 ms from the stimulus onset; from here forward the 'early-delay IEM'; $t_{29} = 2.81$, $p = 0.009$, Cohen's $d = 0.51$; figure 3). This *early-delay IEM* was identified in the *Pilot Study* to be optimal for reconstructing stimulus identity from 2-back data and was used to test all three *principal hypotheses*.

### 3.2.3. Testing principal hypotheses

#### 3.2.3.1. Principal hypothesis 1

Testing an IEM on 2-back data from the 2000 ms centred in the ISI between items $n$ and $n + 1$, when item $n$ is a UMI, produced a reconstruction that was negatively correlated with the basis function used to train the IEM ($t_{29} = -2.84$, $p = 0.016$, Cohen's $d = -0.52$; figure 4). Therefore, the hypothesis that the UMI would take on a format opposite of its 'perceptual representation' was confirmed.

#### 3.2.3.2. Principal hypothesis 2

Testing an IEM on 2-back data from the 2000 ms centred in the ISI between items $n + 1$ and $n + 2$, when item $n$ is a PMI, did not produce a reconstruction that was positively correlated with the basis function used to train the IEM ($t_{29} = 0.98$, $p = 0.335$, Cohen's $d = 0.18$; figure 4). Therefore, the hypothesis that the PMI would take on the same format as its 'perceptual representation' was not confirmed.

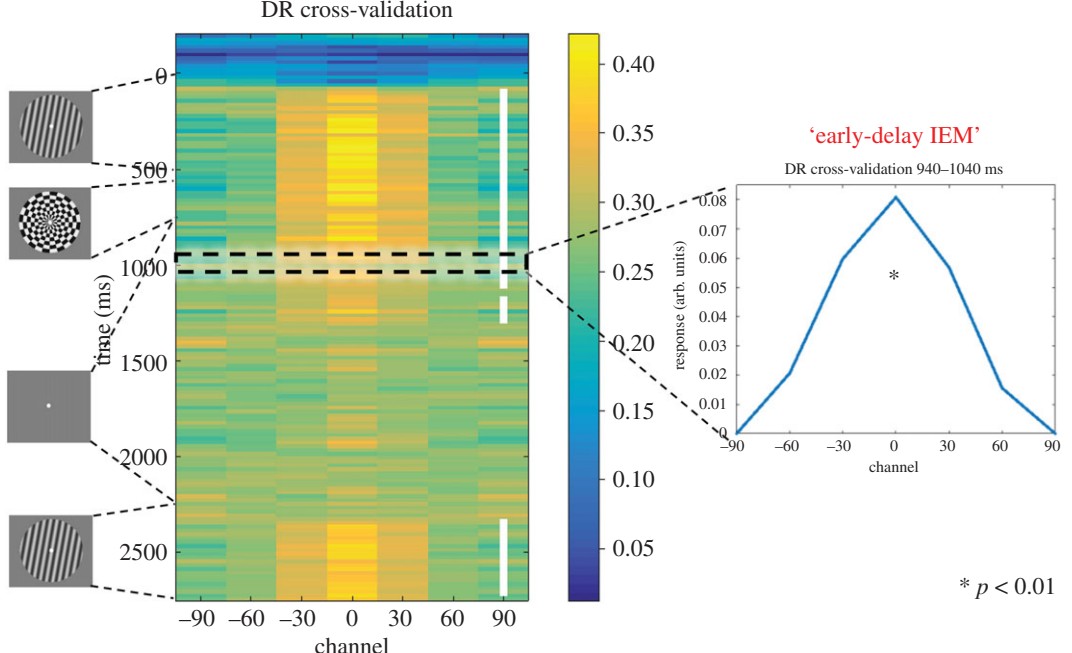

**Figure 3.** Stimulus reconstruction of the delayed-recognition task. Shown is the time course of stimulus reconstruction throughout a delayed-recognition trial (from −200 ms relative to sample onset to probe offset; trained and tested on the same time point) using a *k*-fold cross-validation procedure. White or red squares in the middle of the 90° column mark time points with significant stimulus reconstruction (cluster-based permutation test: $p < 0.05$, two tailed; for all the IEM heatmap figures, we follow the colouring scheme that clusters that have positive *t*-values are coloured white, and clusters that have negative *t*-values are coloured red). On the right is the reconstruction of the early-delay stimulus representation (940–1040 ms from stimulus onset) that was used for reconstruction in N-back tasks.

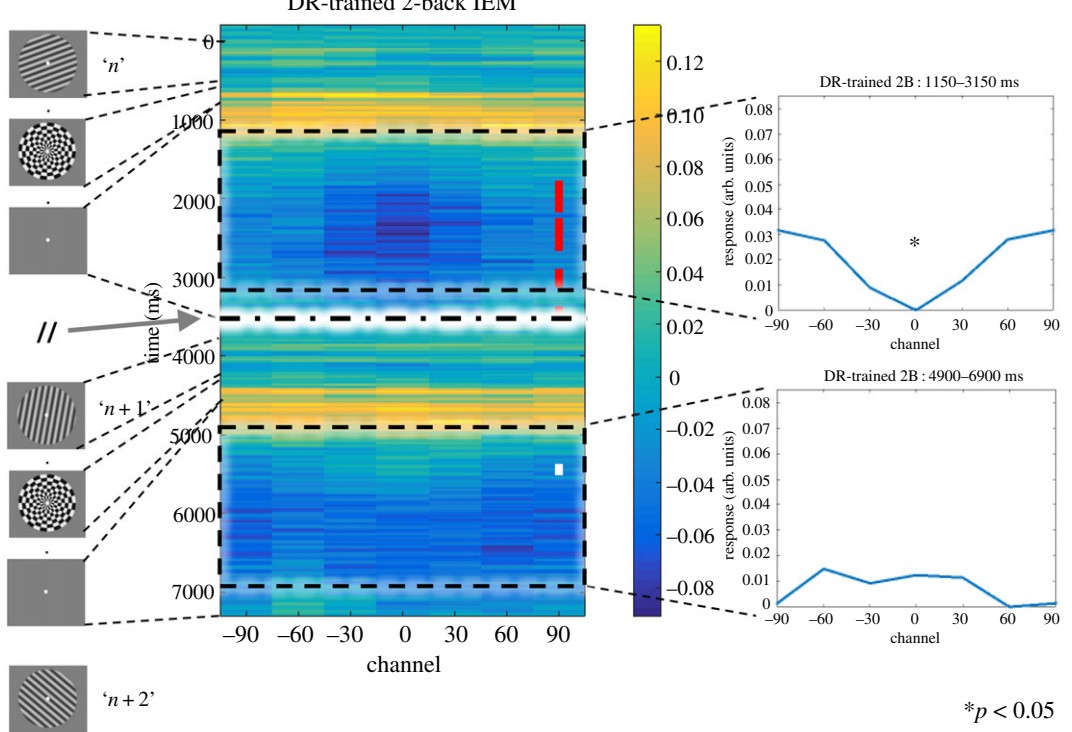

**Figure 4.** Stimulus reconstruction of the 2-back task. Concatenation of the item *n* and item *n* + 1 epochs, each epoch running from 200 ms before stimulus onset to 2.8 s after mask offset. The dashed line in the centre of the figure marks the discontinuity between the two epochs, due to variable-length ISIs. Time-point-by-time-point significance is indicated as shown in figure 3. On the right are reconstructed tuning curves corresponding to the two 2 s windows centred in the post-mask ISIs after item *n* and item *n* + 1, respectively. '*' indicates $p < 0.05$ (two-tailed *t* test), FDR corrected for multiple comparisons.

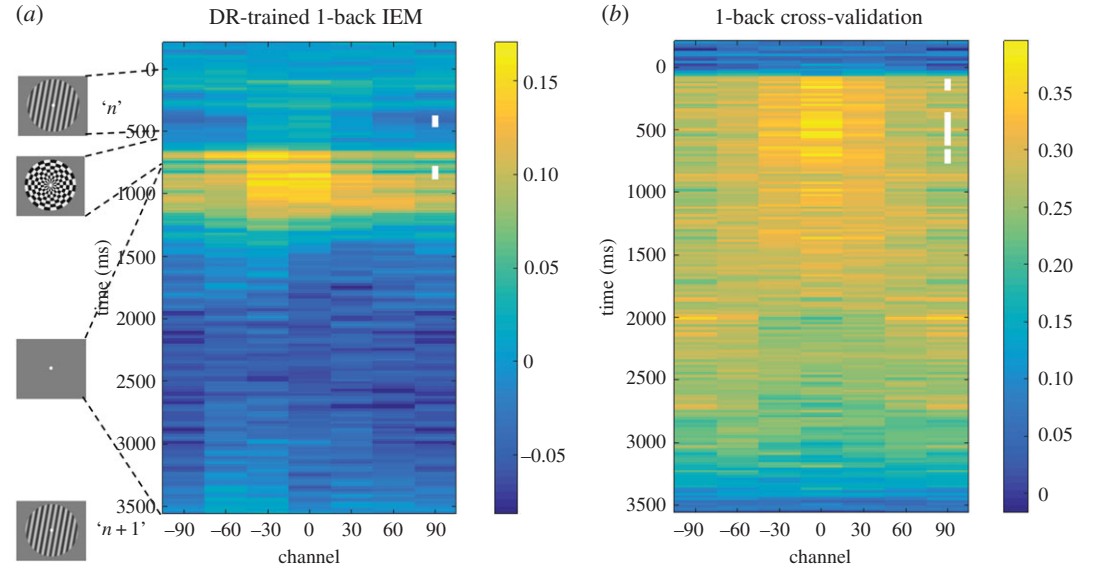

**Figure 5.** Stimulus reconstruction of the 1-back task. (*a*) 1-back reconstruction time course based on the delayed-recognition-trained early-delay IEM. (*b*) 1-back reconstruction time course using *k*-fold cross-validation. Time-point-by-time-point significance is indicated as shown in figure 3.

### 3.2.3.3. Principal hypothesis 3

Testing an IEM on 1-back data from the 2000 ms centred in the ISI between items *n* and *n* + 1, when item *n* is a PMI, did not produce a reconstruction that was positively correlated with the basis function used to train the IEM ($t_{29} = 0.66$, $p = 0.515$, Cohen's $d = 0.12$; figure 5*a*). Therefore, the hypothesis that the priority status of item *n* would result in a reconstruction in its 'perceptual format' was not confirmed. Note, however, that because the IEM reconstruction from this 2000 ms epoch was numerically positive and because brief periods of positive reconstruction were identified in the time-point-by-time-point analysis (figure 5*a*), the substantively more important implication of this result was that it was inconsistent with the possibility that IEM reconstructions that are negatively correlated with the basis function used to train the IEM can be explained by a post-stimulus undershoot in the neural data.

### 3.2.4. Testing secondary hypotheses

### 3.2.4.1. Secondary hypothesis 1

As shown in figure 5, IEM reconstruction results from 1-back are qualitatively similar between those from *k*-fold cross-validation and from testing the *early-delay IEM* trained on the delayed-recognition task: Both revealed significant clusters of positive reconstructions between 0 and 1000 ms, i.e. during stimulus presentation and early delay. This aligns with *secondary hypothesis 1*. Note that stimulus reconstruction was more successful in 1-back task (figure 5) compared with the ISI between items *n* + 1 and *n* + 2 of the 2-back (*principal hypothesis 2*; figure 4). This is probably due to the difference between the two tasks in memory load.

### 3.2.4.2. Secondary hypotheses 2–9

To examine how microsaccades contribute to non-spatial working memory performance, *secondary hypotheses 2–9* aimed to decode stimulus orientation from gaze position from IR-based eye tracking.

In the 2-back task, using *k*-fold cross-validation and a multi-class probabilistic classifier, classification accuracy was not significantly different from chance level (0.1667) when we tried to decode item *n*'s orientation during its presentation ($t_{29} = -0.90$, $p = 0.377$, Cohen's $d = -0.16$; figure 6*a*), an outcome that did not support *secondary hypothesis 2*. We were, however, able to successfully decode the orientation of both item *n* (UMI; $t_{29} = 2.27$, $p = 0.047$, Cohen's $d = 0.41$) and of item *n* − 1 (PMI; $t_{29} = 3.40$, $p = 0.006$, Cohen's $d = 0.62$) during the 2000 ms of the ISI between items *n* and *n* + 1. These results, while numerically very small, are consistent with *secondary hypotheses 3* and *4*, although inspection of figure 6*a* reveals no evidence of switching attention back and forth between the PMI and the UMI (*secondary*

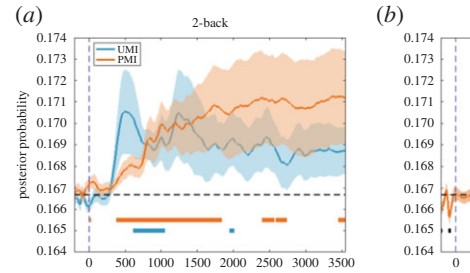
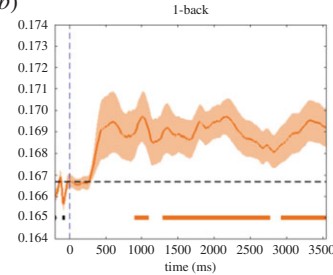
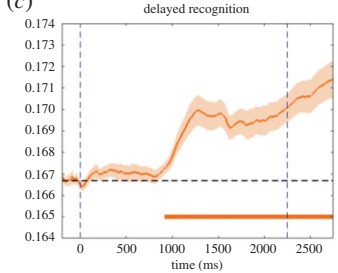

**Figure 6.** Stimulus classification from gaze position data using a multi-class probabilistic classifier and *k*-fold cross-validation. (*a*) 2-back, (*b*) 1-back and (*c*) delayed-recognition classification time course using *k*-fold cross-validation. Bars above the time axis indicate classification significantly different from chance: orange and blue = above-chance classification of the PMI and UMI, respectively; black = significantly below-chance classification.

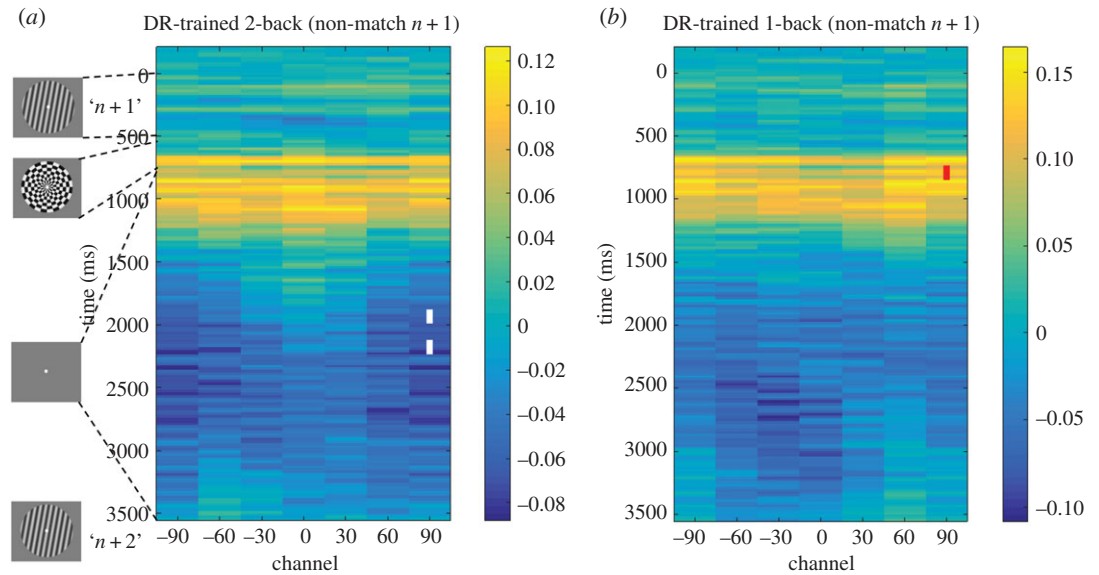

**Figure 7.** Stimulus reconstruction of item *n* time-locked to the presentation of item *n* + 1 during epochs when *n* + 1 was a non-match to *n*, with the delayed-recognition-trained early-delay IEM, during (*a*) 2-back (*secondary hypothesis 10*), and (*b*) 1-back (*secondary hypothesis 11*). Time-point-by-time-point significance is indicated as shown in figure 3.

*hypothesis 5*). Thus, although perception-related eye position does not support the decoding of stimulus identity, eye position during the ISI does support a modest level of decoding of both the PMI and the UMI.

In both the 1-back task and the delayed-recognition task, the results were similar. For 1-back, item *n*'s orientation could not be decoded from its presentation period ($t_{29} = 1.63$, $p = 0.113$, Cohen's $d = 0.30$; failing to support *secondary hypothesis 6*), but it could be decoded successfully during the 2000 ms ISI between items *n* and *n* + 1 ($t_{29} = 2.87$, $p = 0.015$, Cohen's $d = 0.52$; supporting *secondary hypothesis 7*; figure 6*b*). For delayed recognition, stimulus decoding was not successful during stimulus presentation ($t_{29} = 1.37$, $p = 0.180$, Cohen's $d = 0.25$; failing to support *secondary hypothesis 8*), but was successful for the 1000 ms ISI ($t_{29} = 5.05$, $p < 0.001$, Cohen's $d = 0.92$; supporting *secondary hypothesis 9*; figure 6*c*).

### 3.2.4.3. Secondary hypotheses 10 and 11

(Note that when these hypotheses were articulated, we assumed that the delayed-recognition-trained *early-delay IEM* would operationalize a 'perceptual representational format', but results from the exploratory analyses of the functional localizer data suggest that this assumption was incorrect.) Testing the *early-delay IEM* on 2-back data failed to produce significant positive reconstruction for item *n* during the presentation of the non-match item *n* + 1 ($t_{29} = -0.42$, $p = 0.678$, Cohen's $d = -0.08$; a failure to support *secondary hypothesis 10*; figure 7*a*). Testing the *early-delay IEM* on 1-back data failed to produce a significant negative reconstruction of item *n* during the 2000 ms ISI between items *n* + 1 and *n* + 2 ($t_{29} = -1.07$, $p = 0.293$, Cohen's $d = -0.20$; consistent with *secondary hypothesis 11*; figure 7*b*;

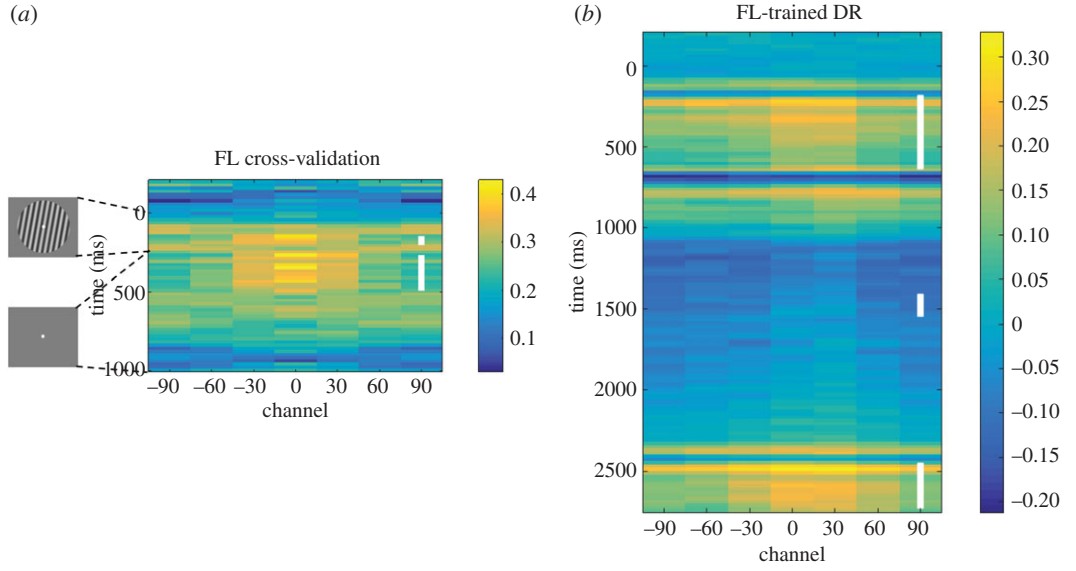

**Figure 8.** Exploratory analyses based on the functional localizer. (*a*) Functional localizer reconstruction time course using *k*-fold cross-validation. (*b*) Delayed-recognition reconstruction time course based on the functional-localizer-trained IEM. Time-point-by-time-point significance is indicated as shown in figure 3.

Note that the brief period of negative-going reconstruction appearing from time 760 to 820 ms in figure 7*b* occurs immediately following the offset of the mask and does not overlap the portion of the ISI when the reconstruction of item *n* is negative during the 2-back task (figure 4)). This latter result serves as evidence against the possible confound that the negative reconstruction of UMI is simply a consequence of an item being superseded in priority, rather than of its deprioritization from a set of competing task-relevant representations.

## 3.3. Exploratory analyses with functional localizer

The *k*-fold cross-validation of IEMs trained on the functional localizer (FL) data yielded clusters of robust reconstruction at the very end of the stimulus presentation epoch (160–180 ms) and during an early portion of the ISI (280–460 ms; figure 8*a*). We chose to train a '*functional localizer IEM*' from the more sustained early-ISI cluster (280–460 ms; $t_{29} = 3.28$, $p = 0.003$, Cohen's $d = 0.60$) to test on the working memory tasks.

### 3.3.1. Delayed recognition

Testing with the *functional localizer IEM* generated robust stimulus reconstruction during the sample and probe periods of the task, when a stimulus was on the screen (figure 8*b*). Notably, however, stimulus reconstruction during the delay period was meagre, extending only 120 ms after sample offset and for a brief period during the delay (1420–1520 ms after trial onset) that did not overlap with the time span of the *early-delay IEM* derived from the delayed-recognition task.

### 3.3.2. N-back tasks

Testing with the *functional localizer IEM* failed to yield any significant reconstructions that were positively correlated with the basis function in either the 1-back or the 2-back task and yielded only one brief period of significant reconstruction of item *n* that was negatively correlated with the basis function during the ISI when *n* was the PMI (1650–1710 ms from *n* + 1 onset; $t_{29} = -2.49$, $p = 0.019$, Cohen's $d = -0.45$; electronic supplementary material, figure S3).

## 4. Discussion

How does the brain keep information 'in mind' when it is outside the focus of attention? Although there are many possibilities for the representation of UMIs, the recent literature has emphasized three: patterns of

synaptic weights that persist after attention is shifted away and elevated activity thus declines (e.g. [3,23,24]); active retention in circuits that are specialized for working memory storage (e.g. [7,25]); and active retention in a representation format that is different from the format of the PMI (e.g. [8,9]). Note that these schemes need not be mutually exclusive, in that, for example, active and synaptic representations can exist at the same time (e.g. [24]), as can multiple active representations (e.g. [7]). In the present study, we have generated evidence consistent with our *principal hypothesis 1*: the UMI is held in an active representation that is transformed relative to when it is a PMI. That is, when an IEM is trained on data corresponding to an item's retention in a single-item delayed-recognition task, that model successfully reconstructs the item from portions of the 2-back task when it is the UMI, but this reconstruction is negatively correlated with the basis function that was used to train the model. When an item is a UMI, it is in a format that is opposite to its format when it is a PMI. This finding is reminiscent of the fMRI finding from van Loon *et al.* [8], in which the representation of the category of the UMI in ventral temporal cortex projected into the opposite location of the MDS space relative to when it was the PMI. In addition, Yu, Teng & Postle [9], in an fMRI study of delayed serial retrocueing (DSR) with oriented gratings, have found IEM evidence for an active-but-opposite representation of the UMI in early visual cortex—i.e. IEM reconstructions of the orientation of the UMI are negatively correlated with the basis function that was used to train the model. Before we consider theoretical implications of our findings, we will review the results of our other preregistered hypotheses.

Our results did not confirm *principal hypothesis 2*, in that we failed to find evidence for the temporally extended representation of the PMI in the 2-back task when testing with an IEM trained on data from the 1-item delayed-recognition task. Of possible relevance is the fact that one difference between the *Pilot Study*, in which the PMI *was* positively reconstructed during the 2-back task, and the *Registered Report* is that in the former, 2-back task blocks were tested consecutively, whereas in the latter, they were interleaved with other tasks, a factor that may have added noise, thereby decreasing sensitivity of the analysis. Regardless of the reason(s), we do not feel that this presents a major problem for the interpretation of the finding of principal theoretical interest—the representational transformation of the UMI—because the trend for the reconstruction of the 2 s 'PMI window' was positive, and the briefer epoch within this window when the PMI was successfully reconstructed with time-point-by-time-point testing yielded positive reconstructions (figure 4; also figure 7*a* in test of *secondary hypothesis 10*). This trend and these significant effects were opposite in sign to the UMI results.

Our results also did not confirm *principal hypothesis 3*, in that they failed to find evidence, from the 1-back task, for sustained delay period representation of the stimulus while it was a PMI. Importantly, however, the 1-back task was included in this experiment to control for an 'undershoot' alternative interpretation of the opposite reconstruction of the UMI during the 2-back task: if the UMI 2-back results were due to a post-stimulus undershoot, rather than to cognitive factors, similarly opposite reconstructions should also be observed for the PMI in the 1-back task, when all perceptual and motoric factors are the same, and only task instructions are different. Instead, the reconstruction of the PMI from the 1-back data was positive and robust while it was on the screen, and this persisted for the first few hundred milliseconds into the delay period (figure 5). Relatedly, the failure to find evidence for an opposite reconstruction of item *n* during the ISI between items *n* + 1 and *n* + 2 of the 1-back task (*secondary hypothesis 11*; figure 7*b*) rule out the possibility that any instance of shifting attention away from an item results in its opposite reconstruction.

*Secondary hypotheses 2–9* addressed the possible contributions of eye position to the results from our encoding analyses of the EEG data. Gaze position analysis revealed that stimulus orientation was decodable from eye position, albeit at extremely low levels of classifier performance, from portions of the delay periods of all three working memory tasks. Interestingly, however, classification was unsuccessful during the stimulus presentation period of these tasks. When we consider the temporal patterns of these results, they do not support the idea that eye position contributed to the results of the EEG analyses in any meaningful way: in the delayed-recognition and 1-back tasks, the onset of successful stimulus reconstruction from the EEG preceded the onset of successful stimulus decoding from eye position; and in the 2-back task, the UMI could be decoded only from eye position early in the delay (plus one later brief blip), whereas the significant opposite IEM reconstruction of the UMI from the EEG data arose later and was sustained (with brief interruptions) until the end of the delay. Nonetheless, the eye position results are consistent with the idea that oculomotor processes may play a functional role in the encoding of non-spatial working memory content [13,26].

The results from the exploratory analyses with the functional localizer task provide a segue for this discussion to return to the topic of primary interest: neural coding during visual working memory. The work presented here is grounded in the rationale that if an IEM trained on data from one condition can

successfully reconstruct stimulus information when tested on data from a different condition, the second condition must engage (at least in part) the same neural code as the first. With respect to the functional localizer task, it was intended to capture the code recruited by the 'passive' visual processing of oriented gratings presented while the subject was attending to a different task. The fact that the functional-localizer-trained IEM reconstructed stimulus orientation from the stimulus-encoding epoch of the delayed-recognition task, but that this ended abruptly with the post-stimulus mask (compare figure 8*b* with figure 3), indicates that the neural code that supports visual working memory is different from a perceptual code that is 'prolonged in time' [27,28]. We can also infer that elements of the mnemonic code that was operationalized by the *early-delay IEM* are stable across time, because of the success of this IEM at reconstructing stimulus information during portions of the 2-back task. Indeed, because it is testing with the *early-delay IEM* that yielded the reconstruction of the UMI that is negatively correlated with the basis function used to train it, we have come to the realization that it is inaccurate to refer to the hypothesized process that produces this result as 'priority-based recoding,' as we did in the Stage 1 write-up of this registered report, because (elements of) the code that supports the UMI must be the same as the one(s) supporting stimulus representation during the early-delay period of the delayed-recognition task. If this were not true, testing with the *early-delay IEM* would fail. For this reason, we believe that 'priority-based remapping' is a better characterization of these results (as well as those of [8,9]): the neural code is the same, but the mappings between specific stimulus values and specific neural patterns have changed. Stated another way, when a stimulus transitions to an unprioritized status, the set of mappings between stimulus values and neural patterns rotates such that the individual mappings are now different, but the distance (in orientation) between neural patterns, and therefore the neural code, is preserved [9]. We tentatively propose that priority-based remapping may be a mechanism that serves to reduce interference among competing task-relevant representations, allowing for 'cleaner' access to the information needed to guide in-the-moment thought and action while also maintaining privileged access to the information that is currently unprioritized.

In the previous work, we have used failures to find evidence for an active representation of the UMI (e.g. [1,3]) to advance the idea that an activity-silent trace may support the retention of the UMI in working memory. Although the results presented here indicate that the UMI *can* be supported by an active neural trace, they do not exclude the possibility that a concurrent activity-silent representation may also support performance (a possibility simulated explicitly by [24]). Indeed, the IEM reconstruction time courses from the delayed-recognition and 1-back tasks indicate that positive reconstructions of the PMI are only sustained during the early portion of the delay period despite high levels of behavioural performance. It seems reasonable to suppose that an activity-silent trace, perhaps supplemented with strategies such as subtle eye movements, might be sufficient for the retention of a single item across an unfilled delay.

In conclusion, the present study provides evidence that UMIs can be represented by an active neural trace, and suggests that a mechanism of priority-based remapping may be deployed to facilitate the flexible guidance of behaviour with the contents of working memory.

Ethics. The experimental protocol, along with the informed consent form, was approved by the University of Wisconsin–Madison Health Institutional Review Board (protocol no. 2016–0500). Prior to each experimental session, informed consent was obtained by laboratory personnel listed on the IRB-approved protocol.

Data accessibility. All raw and processed data, analysis scripts, and the approved Stage 1 protocol for this study are accessible on the Open Science Framework (https://osf.io/yp8h5/; doi:10.17605/OSF.IO/YP8H5).

Authors' contributions. B. R. P., Q. W., Y. C. and J. S. designed the study. Q. W. collected the data. Q. W., Y. C. and J. S. analysed the data. Q.W. and B.R.P. interpreted the results and wrote the manuscript. All authors gave final approval for publication.

Competing interests. The authors have no competing interests.

Funding. This work was supported by the National Institutes of Health (grant no. R01-MH064498).

Acknowledgements. We thank Drs. Yuri Saalmann, Joseph Austerweil, Bas Rokers, Jacqueline Fulvio and Qing Yu for their critical feedback, and Zengbo Xie and Pingyao Wang for their help with data collection.

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
