## [Reviewer comments · Royal Society Open Science]

Review History

Decision letter (RSOS-190061.R0)

16-Jan-2019

Dear Mr Wan,

I write you in regards to manuscript RSOS-190061 entitled "Tracking stimulus representation across a 2-back visual working memory task" which you submitted to Royal Society Open Science.

We routinely triage submissions for scientific soundness, clarity and general adherence to the Registered Reports guidelines. For submissions that have promise but are not yet suitable for in-depth Stage 1 review, we offer feedback to help authors maximise the chances that reviewers will respond positively to a resubmission.

We have concluded that your submission is not yet suitable for in-depth review and has therefore been rejected at this time, but we believe it will be suitable once several issues are addressed. We therefore invite a resubmission. Further comments from the Associate Editor may be found at the end of this letter.

If you wish to revise your manuscript in light of the below comments please submit your

manuscript as a new submission and mention this previous manuscript ID in your covering letter. You should also provide a detailed response to the below comments in the cover letter.

Please note that Royal Society Open Science will introduce article processing charges for all new submissions received from 1 January 2018. Registered Reports submitted and accepted after this date will ONLY be subject to a charge if they subsequently progress to and are accepted as Stage 2 Registered Reports. If your manuscript is submitted and accepted for publication after 1 January 2018 (i.e. as a full Stage 2 Registered Report), you will be asked to pay the article processing charge, unless you request a waiver and this is approved by Royal Society Publishing. You can find out more about the charges at <http://rsos.royalsocietypublishing.org/page/charges>. Should you have any queries, please contact openscience@royalsociety.org.

Thank you for considering Royal Society Open Science for the publication of your registered report.

on behalf of Professor Chris Chambers (Registered Reports Editor, Royal Society Open Science)
openscience@royalsociety.org

Associate Editor Comments to Author:

Should you decide to resubmit, we would be grateful if you could address the following issues:

1. The listing of the primary hypotheses is clear, but please ensure that each hypothesis is associated directly with a specific statistical test (these can also be listed in the analysis section to maximise clarity). The mapping between analysis plans and hypotheses is currently too broad to proceed to in-depth review.
2. Please ensure that all procedures and analysis plans include sufficient detail to be reproducible, without requiring readers to read prior work (of course this prior work can and should be cited, but readers should not need to rely upon it to reproduce the proposed methods). For example, this description of p22: "...we will feed gaze-position data to a multi-class probabilistic classifier to decode stimulus orientation, following the methods of Mostert et al. (2018)" should be replaced with a detailed and reproducible protocol of the procedure.
3. Please provide additional details concerning the power analysis, including the effect size estimate included in each calculation and any additional assumptions. While the inclusion of pilot data is welcome, power analyses should generally not be based solely on the effect size estimate of the pilot study but on the minimal plausible yet theoretically interesting value or the lower bound estimate extracted from a set of previous studies (including the pilot). Basing the power analysis on a single point value is inadvisable because the estimate of the effect size from your pilot study is consistent with a range of values, including effects much smaller than the estimate that may nonetheless be theoretically informative. Moreover, if your estimates from the pilot study are a selection of possible effects from a large pool of tests of the difference between conditions, then they are likely to be over-estimates. If you wish to use the pilot study alone for sample planning then at a minimum we recommend powering your pre-registered study to the lower limit of the confidence interval on the obtained effect size, combined with a rationale for why a lower effect size is not of theoretical interest. This issue is frequently raised during in-depth in statistical review of Stage 1 Registered Reports; therefore it is more efficient to handle it now to expedite the eventual review process.

Author's Response to Decision Letter for (RSOS-190061.R0)

See Appendix A.

RSOS-190228.R0

Review form: Reviewer 1

Is the language acceptable?

Yes

Do you have any ethical concerns with this paper?

No

Have you any concerns about statistical analyses in this paper?

No

Recommendation?

Accept in principle

Comments to the Author(s)

This is an excellent Registered Report. The research question is well motivated both theoretically and empirically. The theoretical motivation is that several theories of working memory (WM) propose a distinction between attended and unattended states of representations within WM. Previous research – not least from the present authors' lab – has provided support for this distinction through MVPA decoding. The empirical motivation arises from the fact that the neural and functional status of unattended information in WM is rather unclear: This information is often not decodable, but occasionally it is, and there are a few intriguing recent instances where it looks as if they are encoded by neural patterns that are the mirror image of the patterns encoding the same information in an attended state. If that finding could be firmly established, it would have important implications for our understanding of WM, and of how the brain represents information it currently deals with: Not-attended information is represented by a pattern of neural activity, but in a different way than attended information. This insight would provide an important lead on how to model WM and attention, and their interplay. The planned study is well designed in all regards – the authors obviously have ample experience with this kind of work, and have given the design and the methods a lot of thought. I found every aspect of the Methods section convincing, and sufficiently detailed to enable others to reproduce the study.

This is a rare case where I review a manuscript and have no complaints at all – but I have three questions/comments that the authors might find useful to further fine-tune their plan:

- The authors say that they do not want to address whether items further back than n-2 disappear from WM through decay or removal, and that's fine. Nevertheless, I think it would be a shame not to use the present data to also investigate the fate of an item's neural representation when it becomes permanently, rather than temporarily, irrelevant. The study of Rose et al. (2016) suggests that these states might differ, in that permanently irrelevant information actually leaves WM. If that is the case, then there should be no neural trace of an item once it recedes beyond the n-2 horizon, as opposed to the mirror-image trace of temporarily-unattended items. Documenting that difference would add value to the study.
- In the Results section the authors identified two time periods during the delayed-matching task that proved "optimal" for decoding during the 2-back task, and they say that they will use the later one (940-1040 ms), but in the Hypothesis section, they speak of an earlier 440-540 ms

window to be used for training the IEM. Which time window will be used? Perhaps more important: Will the time window now be fixed for the preregistered study, or again chosen to be optimal based on the data from that study? The latter may or may not be problematic, depending on how exactly the criterion of optimality is defined – on which unfortunately there was very little information.

- Calculating sample sizes a priori is always risky, especially when venturing into new territory. The authors did a good job in estimating effect sizes and power, but there remains the risk that the study will fall just short of a significant result on one of the main hypotheses. The authors might consider using Bayesian statistics to test the hypotheses. This would enable them to add further subjects to the sample in case the evidence remains ambiguous after $N=30$ (Rouder, 2014). Rouder, J. N. (2014). Optional stopping: No problem for Bayesians. *Psychonomic Bulletin & Review*, 21, 301-308.

Review form: Reviewer 2 (Christian Olivers)

Is the language acceptable?

Yes

Do you have any ethical concerns with this paper?

No

Have you any concerns about statistical analyses in this paper?

No

Recommendation?

Accept in principle

Comments to the Author(s)

RSOS-190228

Tracking stimulus representation across a 2-back visual working memory task

Wan, Quan - University of Wisconsin-Madison

Cai, Ying - Beijing Normal University

Samaha, Jason - University of California, Santa Cruz

Postle, Bradley - University of Wisconsin-Madison

Article Type: Registered Report - Stage 1

Submitted: 06-Feb-2019

Evaluation

I think this is an exemplary registration report. The pilot experiment + data is very useful, and provide a solid basis for the research plan, including the analyses. The proposed changes for the new experiment make total sense. The writing is very clear, the methods appear complete to me. Last but not least, the research question is very topical and exciting (but that might be my bias). So it gets all the green lights from me. I have a few comments/questions. The authors can decide whether they make sense and need to be incorporated. I do not necessarily need to see the response before they go ahead.

1. Is the single item delayed recognition task still necessary if you will also include a 1-back condition in the main experiment? Does the latter not allow for robust training of itself?
2. As an additional exploratory analysis I would find it interesting to see if the N-1 stimulus can be reconstructed from the non-match n stimuli, and what that reconstruction would then look

like. This situation would be close to the resuscitation through unrelated stimulation scenario (even though the mask serves this function better here).

3. In the pilot data, why was behavioural performance in the 2-back better than in the single item delayed recognition task? This was counterintuitive to me. Do the authors expect this to happen again in the experiment proper?

4. The last two p-values on p. 25 seem incorrect (I did not check all of them)

5. A study by Greene et al (2015) appears relevant as background:
Greene, C. M., Kennedy, K., & Soto, D. (2015). Dynamic states in working memory modulate guidance of visual attention: Evidence from an n-back paradigm. *Visual Cognition*, 23(5), 546-560.

signed,
Chris Olivers

Review form: Reviewer 3 (Heleen Slagter)

Is the language acceptable?

Yes

Do you have any ethical concerns with this paper?

No

Have you any concerns about statistical analyses in this paper?

No

Recommendation?

Accept with minor revision

Comments to the Author(s)

See attached file (Appendix B).

Decision letter (RSOS-190228.R0)

18-Mar-2019

Dear Mr Wan

On behalf of the Editors, I am pleased to inform you that your Stage 1 Registered Report RSOS-190228 entitled "Tracking stimulus representation across a 2-back visual working memory task" has been accepted in principle for publication in *Royal Society Open Science* subject to minor revision in accordance with the referee and editor suggestions. Please find their comments at the end of this email.

The reviewers and handling editors have recommended publication, but also suggest some minor revisions to your manuscript. Therefore, I invite you to respond to the comments and revise your manuscript.

To revise your manuscript, log into <https://mc.manuscriptcentral.com/rsos> and enter your

Author Centre, where you will find your manuscript title listed under "Manuscripts with Decisions". Under "Actions," click on "Create a Revision." You will be unable to make your revisions on the originally submitted version of the manuscript. Instead, revise your manuscript and upload a new version through your Author Centre.

Full author guidelines can be found here
<http://rsos.royalsocietypublishing.org/content/registered-reports>.

on behalf of Professor Chris Chambers (Subject Editor, Royal Society Open Science)
openscience@royalsociety.org

Associate Editor Comments to Author (Professor Chris Chambers):

Three expert reviewers have now appraised the manuscript. The assessments are broadly very positive. Each review, however, does raise specific issues that should be addressed before the awarding of Stage 1 acceptance, including potential additional questions that might be asked from this design, provision of additional methodological detail and justification of design designs (including sample size), and potential inclusion of additional procedures (e.g. a localizer task). Please respond to carefully to each point raised.

Reviewer comments to Author:
Reviewer: 1

Comments to the Author(s)

This is an excellent Registered Report. The research question is well motivated both theoretically and empirically. The theoretical motivation is that several theories of working memory (WM) propose a distinction between attended and unattended states of representations within WM. Previous research – not least from the present authors' lab – has provided support for this distinction through MVPA decoding. The empirical motivation arises from the fact that the neural and functional status of unattended information in WM is rather unclear: This information is often not decodable, but occasionally it is, and there are a few intriguing recent instances where it looks as if they are encoded by neural patterns that are the mirror image of the patterns encoding the same information in an attended state. If that finding could be firmly established, it would have important implications for our understanding of WM, and of how the brain represents information it currently deals with: Not-attended information is represented by a pattern of neural activity, but in a different way than attended information. This insight would provide an important lead on how to model WM and attention, and their interplay. The planned study is well designed in all regards – the authors obviously have ample experience with this kind of work, and have given the design and the methods a lot of thought. I found every aspect of the Methods section convincing, and sufficiently detailed to enable others to reproduce the study.

This is a rare case where I review a manuscript and have no complaints at all – but I have three questions/comments that the authors might find useful to further fine-tune their plan:

- The authors say that they do not want to address whether items further back than n-2 disappear from WM through decay or removal, and that's fine. Nevertheless, I think it would be a shame not to use the present data to also investigate the fate of an item's neural representation when it becomes permanently, rather than temporarily, irrelevant. The study of Rose et al. (2016) suggests that these states might differ, in that permanently irrelevant information actually leaves WM. If that is the case, then there should be no neural trace of an item once it recedes beyond the n-2 horizon, as opposed to the mirror-image trace of temporarily-unattended items. Documenting that difference would add value to the study.

- In the Results section the authors identified two time periods during the delayed-matching task that proved "optimal" for decoding during the 2-back task, and they say that they will use the later one (940-1040 ms), but in the Hypothesis section, they speak of an earlier 440-540 ms window to be used for training the IEM. Which time window will be used? Perhaps more important: Will the time window now be fixed for the preregistered study, or again chosen to be optimal based on the data from that study? The latter may or may not be problematic, depending on how exactly the criterion of optimality is defined – on which unfortunately there was very little information.

- Calculating sample sizes a priori is always risky, especially when venturing into new territory. The authors did a good job in estimating effect sizes and power, but there remains the risk that the study will fall just short of a significant result on one of the main hypotheses. The authors might consider using Bayesian statistics to test the hypotheses. This would enable them to add further subjects to the sample in case the evidence remains ambiguous after N=30 (Rouder, 2014). Rouder, J. N. (2014). Optional stopping: No problem for Bayesians. *Psychonomic Bulletin & Review*, 21, 301-308.

Reviewer: 2

Comments to the Author(s)

RSOS-190228

Tracking stimulus representation across a 2-back visual working memory task

Wan, Quan - University of Wisconsin-Madison

Cai, Ying - Beijing Normal University

Samaha, Jason - University of California, Santa Cruz

Postle, Bradley - University of Wisconsin-Madison

Article Type: Registered Report - Stage 1

Submitted: 06-Feb-2019

Evaluation

I think this is an exemplary registration report. The pilot experiment + data is very useful, and provide a solid basis for the research plan, including the analyses. The proposed changes for the new experiment make total sense. The writing is very clear, the methods appear complete to me. Last but not least, the research question is very topical and exciting (but that might be my bias). So it gets all the green lights from me. I have a few comments/questions. The authors can decide whether they make sense and need to be incorporated. I do not necessarily need to see the response before they go ahead.

1. Is the single item delayed recognition task still necessary if you will also include a 1-back condition in the main experiment? Does the latter not allow for robust training of itself?
2. As an additional exploratory analysis I would find it interesting to see if the N-1 stimulus can be reconstructed from the non-match n stimuli, and what that reconstruction would then look like. This situation would be close to the resuscitation through unrelated stimulation scenario (even though the mask serves this function better here).

3. In the pilot data, why was behavioural performance in the 2-back better than in the single item delayed recognition task? This was counterintuitive to me. Do the authors expect this to happen again in the experiment proper?
4. The last two p-values on p. 25 seem incorrect (I did not check all of them)
5. A study by Greene et al (2015) appears relevant as background:
Greene, C. M., Kennedy, K., & Soto, D. (2015). Dynamic states in working memory modulate guidance of visual attention: Evidence from an n-back paradigm. *Visual Cognition*, 23(5), 546-560.

signed,
Chris Olivers

Reviewer: 3

Comments to the Author(s)
See attached file.

Author's Response to Decision Letter for (RSOS-190228.R0)

See Appendix C.

RSOS-190228.R1 (Revision)

Review form: Reviewer 1

Is the language acceptable?

Yes

Do you have any ethical concerns with this paper?

No

Have you any concerns about statistical analyses in this paper?

No

Recommendation?

Accept in principle

Comments to the Author(s)

I already found the first version of this registered report very convincing, and the present revision addresses all the – relatively minor – issues I had raised before. I therefore think that it is time to give the authors the go-ahead for running the planned study.

I have one comment on the matter of preregistered sample sizes, and I recognize that this is not an issue for the present authors to resolve, but rather one to consider for the general policy of preregistration: When researchers plan to use Bayesian inference statistics, preregistering a fixed sample size is unnecessary, and I think it is even irrational because it binds researchers to a stopping rule for data collection that, from a Bayesian perspective, is suboptimal. An optimal

stopping rule is one where data are collected until an evidence criterion (e.g., a Bayes factor > 10 either for or against the null hypothesis). I think it would be reasonable to preregister such a stopping rule rather than a fixed sample size.

Review form: Reviewer 2 (Christian Olivers)

Is the language acceptable?

Yes

Do you have any ethical concerns with this paper?

No

Have you any concerns about statistical analyses in this paper?

No

Recommendation?

Accept in principle

Comments to the Author(s)

The authors have adequately responded to my earlier comments and I recommend proceeding with this proposed work as is.

-CO

Review form: Reviewer 3 (Heleen Slagter)

Is the language acceptable?

Yes

Do you have any ethical concerns with this paper?

No

Have you any concerns about statistical analyses in this paper?

No

Recommendation?

Accept in principle

Comments to the Author(s)

See attached file (Appendix D). I look forward to the results of your study!

Decision letter (RSOS-190228.R1)

14-May-2019

Dear Mr Wan

On behalf of the Editor, I am pleased to inform you that your Manuscript RSOS-190228.R1 entitled "Tracking stimulus representation across a 2-back visual working memory task" has been

accepted in principle for publication in Royal Society Open Science. The reviewers' and editors' comments are included at the end of this email.

You may now progress to Stage 2 and complete the study as approved. Before commencing data collection we ask that you:

- 1) Update the journal office as to the anticipated completion date of your study.
- 2) Register your approved protocol on the Open Science Framework (<https://osf.io/rr>) or other recognised repository, either publicly or privately under embargo until submission of the Stage 2 manuscript. Please note that a time-stamped, independent registration of the protocol is mandatory under journal policy, and manuscripts that do not conform to this requirement cannot be considered at Stage 2. The protocol should be registered unchanged from its current approved state, with the time-stamp preceding implementation of the approved study design.

Following completion of your study, we invite you to resubmit your paper for peer review as a Stage 2 Registered Report. Please note that your manuscript can still be rejected for publication at Stage 2 if the Editors consider any of the following conditions to be met:

- The results were unable to test the authors' proposed hypotheses by failing to meet the approved outcome-neutral criteria.
- The authors altered the Introduction, rationale, or hypotheses, as approved in the Stage 1 submission.
- The authors failed to adhere closely to the registered experimental procedures. Please note that any deviations from the approved experimental procedures must be communicated to the editor immediately for approval, and prior to the completion of data collection. Failure to do so can result in revocation of in-principle acceptance and rejection at Stage 2 (see complete guidelines for further information).
- Any post-hoc (unregistered) analyses were either unjustified, insufficiently caveated, or overly dominant in shaping the authors' conclusions.
- The authors' conclusions were not justified given the data obtained.

We encourage you to read the complete guidelines for authors concerning Stage 2 submissions at <http://rsos.royalsocietypublishing.org/content/registered-reports>. Please especially note the requirements for data sharing, reporting the URL of the independently registered protocol, and that withdrawing your manuscript will result in publication of a Withdrawn Registration.

Please note that Royal Society Open Science will introduce article processing charges for all new submissions received from 1 January 2018. Registered Reports submitted and accepted after this date will ONLY be subject to a charge if they subsequently progress to and are accepted as Stage 2 Registered Reports. If your manuscript is submitted and accepted for publication after 1 January 2018 (i.e. as a full Stage 2 Registered Report), you will be asked to pay the article processing charge, unless you request a waiver and this is approved by Royal Society Publishing. You can find out more about the charges at <http://rsos.royalsocietypublishing.org/page/charges>. Should you have any queries, please contact openscience@royalsociety.org.

Once again, thank you for submitting your manuscript to Royal Society Open Science and we look forward to receiving your Stage 2 submission. If you have any questions at all, please do not hesitate to get in touch. We look forward to hearing from you shortly with the anticipated submission date for your stage two manuscript.

Kind regards,
Andrew Dunn
Royal Society Open Science Editorial Office
Royal Society Open Science

on behalf of Chris Chambers (Registered Reports Editor, Royal Society Open Science)
openscience@royalsociety.org

Reviewers' comments to Author:

Reviewer: 3

Comments to the Author(s)

See attached file. I look forward to the results of your study!

Reviewer: 1

Comments to the Author(s)

I already found the first version of this registered report very convincing, and the present revision addresses all the – relatively minor – issues I had raised before. I therefore think that it is time to give the authors the go-ahead for running the planned study.

I have one comment on the matter of preregistered sample sizes, and I recognize that this is not an issue for the present authors to resolve, but rather one to consider for the general policy of preregistration: When researchers plan to use Bayesian inference statistics, preregistering a fixed sample size is unnecessary, and I think it is even irrational because it binds researchers to a stopping rule for data collection that, from a Bayesian perspective, is suboptimal. An optimal stopping rule is one where data are collected until an evidence criterion (e.g., a Bayes factor > 10 either for or against the null hypothesis). I think it would be reasonable to preregister such a stopping rule rather than a fixed sample size.

Reviewer: 2

Comments to the Author(s)

The authors have adequately responded to my earlier comments and I recommend proceeding with this proposed work as is.

-CO

Author's Response to Decision Letter for (RSOS-190228.R1)

See Appendix E.

RSOS-190228.R2 (Revision)

Review form: Reviewer 1

Is the manuscript scientifically sound in its present form?

Yes

Are the interpretations and conclusions justified by the results?

Yes

Is the language acceptable?

Yes

Do you have any ethical concerns with this paper?

No

Have you any concerns about statistical analyses in this paper?

No

Recommendation?

Accept with minor revision

Comments to the Author(s)

This is an excellent manuscript. The authors carefully carried out the registered experiment, made their hypotheses explicit, and reported the relevant results in a transparent manner. The discussion is reasonable and the conclusion convincing. I have only a few comments on the writing that I hope will help the authors to improve the clarity of this report:

(1) When presenting their hypotheses, the authors repeat after each hypothesis "The precise method we will use..." - it is enough to say this one (for all hypotheses). It is tiring to read this over and over again.

(2) Secondary Hypothesis 10 is described in the text as reconstruction of item n-1 during presentation of non-matching item n. In Figure 7a, the relevant results are shown as reconstruction of item n during presentation of non-matching item n+1. This is the same, of course, but the mismatch in labeling is confusing. Perhaps more important: How do the data presented in Figure 7a differ from the data in the second period (after onset of stimulus n+1) in Figure 4? And related, how does Principal H 2 differ from Secondary H 10?

(3) On p. 22, should the power of 6 not be added after "(x)"?

(4) On p. 27, the first equation needs a closing parenthesis after x.

Review form: Reviewer 2 (Christian Olivers)**Is the manuscript scientifically sound in its present form?**

Yes

Are the interpretations and conclusions justified by the results?

Yes

Is the language acceptable?

Yes

Do you have any ethical concerns with this paper?

No

Have you any concerns about statistical analyses in this paper?

No

Recommendation?

Accept with minor revision

Comments to the Author(s)

Review of RSOS-190228.R2 Wan, Cai, Samaha, & Postle, "Tracking stimulus representation across a 2-back visual working memory task", as submitted to Royal Society Open Science

I reviewed the preregistered report for this study, and I think it has worked out really nicely. Not all hypotheses have been confirmed, but that would have been a stretch, and the exact reason why a preregistration is so nice. The essential predictions were confirmed, making this study very relevant. I recommend publication, and I congratulate the authors with a very nice study.

I do have a few remarks/questions. I do not need to see the paper again, but the authors may take these along in a final version.

In the Discussion (see also Hypothesis 1) it is argued that the unprioritized memory (UMI) is held in an active representation. By that I assume a patterns of neural firing activity. This a) is a bit odd, since later on in the discussion it is argued that the prioritized memory (PMI) is represented silently; and b) strikes me as unnecessary, as the UMI could still be encoded in connectivity pattern, and what is being picked up is random or nonspecific activity flowing through the network? In any case, unless I'm missing the logic, I think the authors would do well to elaborate on why they assume the UMI to be active, and the PMI passive...

On p. 38 it is argued that instead of re-coding, we better think of it as re-mapping. However, I found it difficult to follow the argument here. That may well be me being thick, but if the authors see a way of clarifying their argument further that would be great.

Minor things:

p.4, bottom: "reconstructed from signal from early visual cortex"

p.6, middle "quality of the task that make it well-suited"

p.23 the conversion of eye x,y coordinates to "visual angle" is not clear. To visual angle as in distance, or to polar coordinates? And if the latter, then only angle or also eccentricity? And if polar angle only, how can we have outliers then? And if it's been converted to simply visual angle (as in distance), it's unclear why that adds anything to the analyses.

Signed,
Chris Olivers

Review form: Reviewer 3 (Heleen Slagter)

Is the manuscript scientifically sound in its present form?

Yes

Are the interpretations and conclusions justified by the results?

Yes

Is the language acceptable?

Yes

Do you have any ethical concerns with this paper?

No

Have you any concerns about statistical analyses in this paper?

No

Recommendation?

Accept with minor revision

Comments to the Author(s)

This is a well written and carefully reasoned paper, providing results from a preregistered EEG study addressing a key outstanding question: How does the neural representation of visual working memory content vary with behavioral priority? The finding that items in working memory that are currently unattended are encoded in an opposite/negative representational format is important as it suggests that priority-based remapping helps to protect this remembered information when it is not yet relevant to the task at hand.

I can confirm that the introduction, rationale and stated hypotheses are the same as the approved Stage1 submission and that the authors adhered precisely to the registered experimental procedures and analyses. The exploratory functional-localizer based analyses are justified, sound, and informative. The authors' conclusions are justified given the data. The observation that an unprioritized item in working memory can be encoded in a distinctive "opposite" representational format compared to a prioritized item provides an important contribution to the rapidly growing literature on the neural mechanisms underlying prioritization of information in working memory.

I only have several more minor comments that the authors might wish to consider.

1. The main finding is that the UMI is associated with negative reconstruction of orientation (Fig. 4), and that this cannot be explained by a post-stimulus undershoot, as this effect is not observed in the 1-back task (Fig. 5a). I did not think of this during the Stage1 review, but it seems critical to me to directly statistically contrast these two effects. Is the reconstruction of the UMI orientation in the 2-back task statistically different from that in the same period in the 1-back task? Although an exploratory analysis at this point, this could strengthen this theoretically interesting finding.

2. A surprising aspect of the current findings is the lack of robust decoding of the PMI in the 2-back task. I think it is important to discuss potential explanations for this unexpected observation, even if speculative.

One possibility is that an item is reprioritized in a different format/neural code (e.g., a verbal vs. visual code). An often-used strategy in n-back tasks in particular at higher levels of n is to verbally rehearse the items to be kept in WM. Although orientation is less easily verbalized than for example letter identity, given that only 6 widely-spread orientations were used, it is possible to verbally label them (e.g., they could be seen as pointing to a position on a clock (e.g., 2 o'clock - > "two")). Could one explanation for some of the current findings then be that (some) participants (at least on some portion of trials) recoded the orientation of the memory items into a verbal format during the 2-back task? I don't readily see how verbal recoding could have led to the observed negative reconstruction of the UIM, but item n may have been recoded to a verbal format after deprioritization to prevent interference from new visual input. This could have subsequently reduced the ability to reconstruct the item when reprioritized in the 2-back task, and could possibly also explain the lack of transfer from the functional localizer trained model to the 2-back task.

Another possibility is that the PMI is more "action-oriented" in nature than (some of) the representations the EIM models were trained on, and also includes contributions from more frontal regions (Myers et al., TiCS, 2017).

A final possibility is that the number of neurons coding the reprioritized PMI is much smaller than the original population used for initial encoding/representation, and hence that there was an active PMI trace, but it was beyond the detection threshold of non-invasive techniques such as EEG. Invasive recordings in humans support this possibility (Kornblith et al., Current Biology, 2017).

3. There is also monkey work showing that some neurons in inferior temporal cortex and V1 remain persistently active during VSTM maintenance (Supèr & Ran, 2008; Supèr et al., 2001; Woloszyn & Sheinberg, JoN, 2009).

4. A study by Olivers and colleagues (De Vries, I. et al., *Neuroimage*, 2019) previously used MVPA to decode memory status (current vs. prospective) from the pattern of scalp EEG activity during a working memory task. This study and its findings seem relevant.

5. This is more of a side thought that may be important for future studies: a grating at fixation covers four quadrants of the visual hemifield, which given the anatomy of V1 (calcarine sulcus) could lead to a very diffuse projection of V1 activity at the level of the scalp. Not ideal for detecting weak effects. (see e.g., Fig 1 in Vagenas, MI et al., *Journal of Neural Engineering*, 2013)

6. For completeness, Figure 8 should also show the results of the FL-trained IEM results for the 2-back task data.

7. Page 5: The abbreviation AMI is used twice. This should be PMI.

8. The color coding used for the figures 3-5, 7-8 is somewhat confusing. Blue colors can indicate both positive and negative reconstruction. Moreover, which color denotes 0 and hence which colors signal positive or negative correlations, changes from figure to figure (e.g., in Figure 3, 0 = blue; in Figure 4, 0 is more greenish-blue; and in Figure 5b, 0 is dark blue). Please make the color coding consistent across figures and so that e.g., green/blue = negative values and yellow/red = positive values.

Decision letter (RSOS-190228.R2)

Dear Mr Wan:

On behalf of the Editor, I am pleased to inform you that your Stage 2 Registered Report RSOS-190228.R2 entitled "Tracking stimulus representation across a 2-back visual working memory task" has been deemed suitable for publication in Royal Society Open Science subject to minor revision in accordance with the referee suggestions. Please find the referees' comments at the end of this email.

The reviewers and Subject Editor have recommended publication, but also suggest some minor revisions to your manuscript. Therefore, I invite you to respond to the comments and revise your manuscript.

Please also ensure that all the below editorial sections are included where appropriate -- if any section is not applicable to your manuscript, please can we ask you to nevertheless include the heading, but explicitly state that the heading is inapplicable. An example of these sections is attached with this email.

- Ethics statement

- Data accessibility

It is a condition of publication that all supporting data are made available either as supplementary information or preferably in a suitable permanent repository. The data

accessibility section should state where the article's supporting data can be accessed. This section should also include details, where possible of where to access other relevant research materials such as statistical tools, protocols, software etc can be accessed. If the data has been deposited in an external repository this section should list the database, accession number and link to the DOI for all data from the article that has been made publicly available. Data sets that have been deposited in an external repository and have a DOI should also be appropriately cited in the manuscript and included in the reference list.

If you wish to submit your supporting data or code to Dryad (<http://datadryad.org/>), or modify your current submission to dryad, please use the following link:
[http://datadryad.org/submit?journalID=RSOS&manu=\(Document not available\)](http://datadryad.org/submit?journalID=RSOS&manu=(Document not available))

- **Competing interests**

- **Authors' contributions**

- **Acknowledgements**

- **Funding statement**

Because the schedule for publication is very tight, it is a condition of publication that you submit the revised version of your manuscript within 7 days (i.e. by the 09-Jul-2020). If you do not think you will be able to meet this date please let me know immediately.

Please note that Royal Society Open Science will introduce article processing charges for all new submissions received from 1 January 2018. Registered Reports submitted and accepted after this date will ONLY be subject to a charge if they subsequently progress to and are accepted as Stage 2 Registered Reports. If your manuscript is submitted and accepted for publication after 1 January 2018 (i.e. as a full Stage 2 Registered Report), you will be asked to pay the article processing charge, unless you request a waiver and this is approved by Royal Society Publishing. You can find out more about the charges at <https://royalsocietypublishing.org/rsos/charges>. Should you have any queries, please contact openscience@royalsociety.org.

on behalf of Professor Chris Chambers
(Registered Reports Editor, Royal Society Open Science)
openscience@royalsociety.org

Associate Editor Comments to Author (Professor Chris Chambers):

Associate Editor: 1

Comments to the Author:

The Stage 2 manuscript was reviewed by the same three expert reviewers who assessed the Stage 1 manuscript. Happily, all are positive about the completed article while also offering valuable suggestions for strengthening the Discussion and clarifying specific aspects of the presentation. Provided the authors are able to respond thoroughly to these points, final Stage 2 acceptance should be forthcoming without requiring further in-depth review.

Comments to Author:

Reviewer: 1

Comments to the Author(s)

This is an excellent manuscript. The authors carefully carried out the registered experiment, made their hypotheses explicit, and reported the relevant results in a transparent manner. The discussion is reasonable and the conclusion convincing. I have only a few comments on the writing that I hope will help the authors to improve the clarity of this report:

(1) When presenting their hypotheses, the authors repeat after each hypothesis "The precise method we will use..." - it is enough to say this one (for all hypotheses). It is tiring to read this over and over again.

(2) Secondary Hypothesis 10 is described in the text as reconstruction of item n-1 during presentation of non-matching item n. In Figure 7a, the relevant results are shown as reconstruction of item n during presentation of non-matching item n+1. This is the same, of course, but the mismatch in labeling is confusing. Perhaps more important: How do the data presented in Figure 7a differ from the data in the second period (after onset of stimulus n+1) in Figure 4? And related, how does Principal H 2 differ from Secondary H 10?

(3) On p. 22, should the power of 6 not be added after "(x)"?

(4) On p. 27, the first equation needs a closing parenthesis after x.

Reviewer: 2

Comments to the Author(s)

Review of RSOS-190228.R2 Wan, Cai, Samaha, & Postle, "Tracking stimulus representation across a 2-back visual working memory task", as submitted to Royal Society Open Science

I reviewed the preregistered report for this study, and I think it has worked out really nicely. Not all hypotheses have been confirmed, but that would have been a stretch, and the exact reason why a preregistration is so nice. The essential predictions were confirmed, making this study very relevant. I recommend publication, and I congratulate the authors with a very nice study.

I do have a few remarks/questions. I do not need to see the paper again, but the authors may take these along in a final version.

In the Discussion (see also Hypothesis 1) it is argued that the unprioritized memory (UMI) is held in an active representation. By that I assume a patterns of neural firing activity. This a) is a bit odd, since later on in the discussion it is argued that the prioritized memory (PMI) is represented silently; and b) strikes me as unnecessary, as the UMI could still be encoded in connectivity pattern, and what is being picked up is random or nonspecific activity flowing through the network? In any case, unless I'm missing the logic, I think the authors would do well to elaborate on why they assume the UMI to be active, and the PMI passive...

On p. 38 it is argued that instead of re-coding, we better think of it as re-mapping. However, I found it difficult to follow the argument here. That may well be me being thick, but if the authors see a way of clarifying their argument further that would be great.

Minor things:

p.4, bottom: "reconstructed from signal from early visual cortex"

p.6, middle "quality of the task that make it well-suited"

p.23 the conversion of eye x,y coordinates to "visual angle" is not clear. To visual angle as in distance, or to polar coordinates? And if the latter, then only angle or also eccentricity? And if polar angle only, how can we have outliers then? And if it's been converted to simply visual angle (as in distance), it's unclear why that adds anything to the analyses.

Signed,
Chris Olivers

Reviewer: 3

Comments to the Author(s)

This is a well written and carefully reasoned paper, providing results from a preregistered EEG study addressing a key outstanding question: How does the neural representation of visual working memory content vary with behavioral priority? The finding that items in working memory that are currently unattended are encoded in an opposite/negative representational format is important as it suggests that priority-based remapping helps to protect this remembered information when it is not yet relevant to the task at hand.

I can confirm that the introduction, rationale and stated hypotheses are the same as the approved Stage1 submission and that the authors adhered precisely to the registered experimental procedures and analyses. The exploratory functional-localizer based analyses are justified, sound, and informative. The authors' conclusions are justified given the data. The observation that an unprioritized item in working memory can be encoded in a distinctive "opposite" representational format compared to a prioritized item provides an important contribution to the rapidly growing literature on the neural mechanisms underlying prioritization of information in working memory.

I only have several more minor comments that the authors might wish to consider.

1. The main finding is that the UMI is associated with negative reconstruction of orientation (Fig. 4), and that this cannot be explained by a post-stimulus undershoot, as this effect is not observed in the 1-back task (Fig. 5a). I did not think of this during the Stage1 review, but it seems critical to me to directly statistically contrast these two effects. Is the reconstruction of the UMI orientation in the 2-back task statistically different from that in the same period in the 1-back task? Although an exploratory analysis at this point, this could strengthen this theoretically interesting finding.

2. A surprising aspect of the current findings is the lack of robust decoding of the PMI in the 2-back task. I think it is important to discuss potential explanations for this unexpected observation, even if speculative.

One possibility is that an item is reprioritized in a different format/neural code (e.g., a verbal vs. visual code). An often-used strategy in n-back tasks in particular at higher levels of n is to verbally rehearse the items to be kept in WM. Although orientation is less easily verbalized than for example letter identity, given that only 6 widely-spread orientations were used, it is possible to verbally label them (e.g., they could be seen as pointing to a position on a clock (e.g., 2 o'clock - > "two")). Could one explanation for some of the current findings then be that (some) participants (at least on some portion of trials) recoded the orientation of the memory items into a verbal format during the 2-back task? I don't readily see how verbal recoding could have led to the observed negative reconstruction of the UIM, but item n may have been recoded to a verbal format after deprioritization to prevent interference from new visual input. This could have subsequently reduced the ability to reconstruct the item when reprioritized in the 2-back task, and could possibly also explain the lack of transfer from the functional localizer trained model to the 2-back task.

Another possibility is that the PMI is more "action-oriented" in nature than (some of) the representations the EIM models were trained on, and also includes contributions from more frontal regions (Myers et al., TiCS, 2017).

A final possibility is that the number of neurons coding the reprioritized PMI is much smaller than the original population used for initial encoding/representation, and hence that there was an active PMI trace, but it was beyond the detection threshold of non-invasive techniques such as EEG. Invasive recordings in humans support this possibility (Kornblith et al., Current Biology, 2017).

3. There is also monkey work showing that some neurons in inferior temporal cortex and V1 remain persistently active during VSTM maintenance (Supèr & Ran, 2008; Supèr et al., 2001; Woloszyn & Sheinberg, JoN, 2009).
4. A study by Olivers and colleagues (De Vries, I. et al., Neuroimage, 2019) previously used MVPA to decode memory status (current vs. prospective) from the pattern of scalp EEG activity during a working memory task. This study and its findings seem relevant.
5. This is more of a side thought that may be important for future studies: a grating at fixation covers four quadrants of the visual hemifield, which given the anatomy of V1 (calcarine sulcus) could lead to a very diffuse projection of V1 activity at the level of the scalp. Not ideal for detecting weak effects. (see e.g., Fig 1 in Vagenas, MI et al., Journal of Neural Engineering, 2013)
6. For completeness, Figure 8 should also show the results of the FL-trained IEM results for the 2-back task data.
7. Page 5: The abbreviation AMI is used twice. This should be PMI.
8. The color coding used for the figures 3-5, 7-8 is somewhat confusing. Blue colors can indicate both positive and negative reconstruction. Moreover, which color denotes 0 and hence which colors signal positive or negative correlations, changes from figure to figure (e.g., in Figure 3, 0 = blue; in Figure 4, 0 is more greenish-blue; and in Figure 5b, 0 is dark blue). Please make the color coding consistent across figures and so that e.g., green/blue = negative values and yellow/red = positive values.

Author's Response to Decision Letter for (RSOS-190228.R2)

See Appendix F.

Decision letter (RSOS-190228.R3)

Dear Mr Wan:

It is a pleasure to accept your Stage 2 Registered Report entitled "Tracking stimulus representation across a 2-back visual working memory task" in its current form for publication in Royal Society Open Science.

on behalf of Professor Chris Chambers (Subject Editor)
openscience@royalsociety.org

Appendix A

6 February 2019

Professor Chris Chambers, Registered Reports Editor
Royal Society Open Science

Dear Chris:

Thank you for looking over our initial submission of the manuscript RSOS-190061 entitled “Tracking stimulus representation across a 2-back visual working memory task,” by Wan, Cai, Samaha, and Postle, and suggesting improvements before it goes to in-depth review. To reaffirm what was stated in the cover letter of 10 January 2019, this submission is appropriate as a Registered Report because it proposes to test a novel hypothesis, one with potentially considerable importance for the cognitive neuroscience of working memory, with a set of quantitative predictions derived from an exploratory, pilot study. What we propose to do here is collect an appropriately powered data set (see below) with a *de novo* sample of subjects, using procedures derived and refined from those that generated preliminary evidence for a mechanism of *priority-based recoding* of information in visual working memory.

We are ready to begin data collection for this study immediately upon receiving Stage 1 *in principle acceptance*, and we anticipate completing data collection within 6 months, and completion of analysis and write-up within 12 months of this date. All necessary support and approvals are in place for the proposed work.

Following Stage 1 *in principle acceptance*, we agree to register our approved protocol on the Open Science Framework.

We agree to share the raw data for all published results.

We confirm that, if we withdraw our paper after provisional acceptance, we agree to the journal publishing a short summary of the pre-registered study under a section Withdrawn Registrations.

On the following pages we specify how this submission was modified in response to your action letter of 16-Jan-2019.

With thanks in advance for your consideration of this submission, and on behalf of my coauthors,

Bradley R. Postle, PhD

In response to action letter of 16-Jan-2019, 'Associate Editor Comments to Author' are reproduced in this sans serif font; and our responses interleaved in this *serifed and italicized Times New Roman font*:

1. The listing of the primary hypotheses is clear, but please ensure that each hypothesis is associated directly with a specific statistical test (these can also be listed in the analysis section to maximise clarity). The mapping between analysis plans and hypotheses is currently too broad to proceed to in-depth review.

We now specify, for each of the principal hypotheses, the sub-section in the Methods that describes the specific statistical test that will used to test it.

2. Please ensure that all procedures and analysis plans include sufficient detail to be reproducible, without requiring readers to read prior work (of course this prior work can and should be cited, but readers should not need to rely upon it to reproduce the proposed methods). For example, this description of p22: "...we will feed gaze-position data to a multi-class probabilistic classifier to decode stimulus orientation, following the methods of Mostert et al. (2018)" should be replaced with a detailed and reproducible protocol of the procedure.

We have made sure all procedures and analysis plans have sufficient detail to be reproducible without requiring readers to refer to previous work. Specifically, we added necessary methodological detail for the multi-class probabilistic classifier employed in Mostert et al. (2018) to decode orientation from gaze position.

3. Please provide additional details concerning the power analysis, including the effect size estimate included in each calculation and any additional assumptions. While the inclusion of pilot data is welcome, power analyses should generally not be based solely on the effect size estimate of the pilot study but on the minimal plausible yet theoretically interesting value or the lower bound estimate extracted from a set of previous studies (including the pilot). Basing the power analysis on a single point value is inadvisable because the estimate of the effect size from your pilot study is consistent with a range of values, including effects much smaller than the estimate that may nonetheless be theoretically informative. Moreover, if your estimates from the pilot study are a selection of possible effects from a large pool of tests of the difference between conditions, then they are likely to be over-estimates. If you wish to use the pilot study alone for sample planning then at a minimum we recommend powering your pre-registered study to the lower limit of the confidence interval on the obtained effect size, combined with a rationale for why a lower effect size is not of theoretical interest. This issue is frequently raised during in-depth in statistical review of Stage 1 Registered Reports; therefore it is more efficient to handle it now to expedite the eventual review process.

We believe that the manuscript now adheres to the spirit of these requests. Because the motivation for this Registered Report is explicitly to replicate the findings from pilot study (and to add a control task), it makes sense to draw on it for the power estimation for the Registered Report. We do appreciate, however, the concerns about basing the power analyses solely on the pilot data, and so have drawn on additional data sets from our lab. (It is the case that this work is, to our knowledge, truly breaking new ground, and so there's nothing that we know of by way of precedent that's already out in the literature.) One of these data sets is an fMRI study from our group that also applies IEM of working memory for line orientation, from which we have computed estimates of both positive and negative IEM reconstructions (these came from the same sample, but different sets of trials). The second is an EEG

study of spatial covert attention – a different class of behavior from the present Registered Report – that we analyzed with IEM – same type of data and same type of analysis. We provide the effect size estimate for each of these sets of results, and the α level that is planned for each statistical test. As you'll see, it turns out, among all these, the pilot data set still yields the largest N, and so that remains what we propose to use, albeit now with a stronger rationale.

Appendix B

The proposed study by Wan et al. examines an important outstanding question: how does behavioral priority affect the representation of information in working memory? To address this question, a recently developed state-of-the-art technique, inverted encoding, trained on independent task data, will be used to reconstruct the to-be-remembered orientation from the pattern of scalp EEG data during a 2-back WM task during two states of behavioral priority (relevant later, relevant now). Moreover, eye tracking will be done concurrently to assess whether the orientation of information being encoded or maintained in WM (whether prioritized or not) can be decoded on the basis of eye position alone. The methods and analyses proposed are generally sound, and the results are certainly going to be of interest to the field. I have two more major suggestions that could strengthen this already strong proposal, next to some more minor suggestions for clarification/improvement.

1. Pilot data on which the preregistered report is based, shown in Figure 4, indicate that contrary to the default assumption that the activity representing an item in working memory might simply get weaker when it is deprioritized, the item is recoded in a different activity pattern, that may (as the authors propose) suggest that a process of *priority-based recoding* helps to protect remembered information when it is not in the focus of attention. Specifically, deprioritization was associated with negative/shifted IEM reconstruction. Yet, this negative/shifted orientation encoding is shown while another representation (that of n-1) is prioritized. In the new study, to more conclusively show that the negative/shifted tuning is indeed related to deprioritization/recoding of item n (rather than another orientation being represented in the activity pattern), a control analysis is necessary. One possibility is to conduct a control analysis in which the IEM is (i) trained on trial n in the 1-back task, i.e., in the absence of any deprioritization of a previously presented item, and (ii) tested on n-1 pretending it is trial n (one could also train on all even trials and test on all uneven trials with the labels of the even trials). One would expect no opposite reconstruction in this case (new hypothesis) IF the observed pattern in the 2-back task is truly related to deprioritization¹. This is important as it has alternatively been suggested that deprioritized items may not be encoded in neural activity patterns, but rather in synaptic efficiency (as the authors also discuss in the introduction).

2. The authors propose to examine to what extent eye position can be used to determine the content of WM, which is important, as a previous study showed that orientation-dependent eye movements may have a systematic effect on the decoded signal (Mostert et al.). Yet, if this truly is the case also in the present study, this could provide a confound to the interpretation of the results of the EEG IEM analyses aimed at reconstructing orientation from the pattern of brain activity. As Mostert et al. pointed out: "If the eyes move, then the projection falling on the retina will also change, even when external visual stimulation remains identical. Thus, if gaze position is systematically modulated by the image that is perceived or kept in mind, then so is the visual information transmitted to the visual cortex. For example, if a vertical grating is presented and

¹ One possibility is that this control analysis of the 1-back data will actually reveal some reconstruction of the non-presented orientation at trial n-1, given that it was recently shown by Bae and Luck (in press, *Psych Science*) that an activity-silent representation of the previous trial is reactivated when the current trial begins. Yet, importantly, this should result in a positive (not negative) reconstruction.

kept in VWM, then the subject may subtly move her or his gaze upward. Correspondingly, the fixation dot is now slightly below fixation, thus leading to visual cortex activity that is directly related to the retinotopic position of the fixation dot.”

In other words, the IEM model may pick up on subtle differences in spatial representation (related to “attending away”), not orientation/priority-based recoding per se, potentially leading to an incorrect conclusion. Mostert et al. also propose a solution: the use of a localizer (training) task that is specifically sensitive to the neural representations encoded in bottom-up signals evoked by passively perceived gratings. Given that Wan et al.’s own pilot data also suggested that the neural code for the perceptual representation of their stimuli is the same as that for their retention in visual working memory, adding such a localizer task to the current design could provide a trained IEM that is not confounded by small eye movements/spatial representation/attending away, and allow for drawing conclusions at the level of orientation recoding in WM. Please consider adding a localizer task in which orientation is not task-relevant.

3) Some important details about the analyses are missing.

a) The authors do not specify if they will filter their EEG data or deal with slow drifts in the data in some other way. A study by van Driel et al. that just came out as a preprint indicates that high-pass filtering can result in temporal displacement of decoding accuracy/information, and propose robust detrending as an alternative. See <https://www.biorxiv.org/content/10.1101/530220v1> Please add information about filter settings/detrending.

b) It is not specified if malfunctioning electrodes will be removed and reinterpolated, and if so, how.

c) It is not clear how trials with eye movements will be identified based on the EEG data: using the EOG traces or ICA components? And why not simply use the more accurate eye tracker data? Will trials with eye blinks during the delay period also be removed from the analysis?

d) Are only correct trials included in the training and reconstruction or are incorrect trials also included in the analyses? Please specify.

e) It is not specified if datasets (subjects) will be excluded from the analyses based on their performance (e.g., when performing x stdev from the group average) or when too few trials are left in a given condition after cleaning. Please add criteria for exclusion if appropriate.

f) Sample size is now specified as $n=30$ based on power analyses/previous work. Does this mean data collection will continue until 30 usable datasets can be included in the analyses (i.e., will bad subjects be replaced or not; or only when less than x good datasets are left)? Please clarify.

4) Recent work by Parthasarathy et al., suggesting code morphing in WM, not cited in the proposal, is relevant to the current ideas.

Parthasarathy, A., Herikstad, R., Bong, J. H., Medina, F. S., Libedinsky, C. & Yen, S. C. Mixed selectivity morphs population codes in prefrontal cortex. *Nature Neuroscience* 20, 1770–1779 (2017)

Parthasarathy, A et al. (bioRxiv). Time-Invariant Working Memory Representations in the Presence of Code-Morphing in the Lateral Prefrontal Cortex.

<https://www.biorxiv.org/content/10.1101/563668v1>

Appendix C

In response to decision letter of 18-Mar-2019, the reviewers' comments are reproduced in this Arial font; and our responses interleaved in this *italicized Times New Roman font*:

Reviewer 1:

- The authors say that they do not want to address whether items further back than n-2 disappear from WM through decay or removal, and that's fine. Nevertheless, I think it would be a shame not to use the present data to also investigate the fate of an item's neural representation when it becomes permanently, rather than temporarily, irrelevant. The study of Rose et al. (2016) suggests that these states might differ, in that permanently irrelevant information actually leaves WM. If that is the case, then there should be no neural trace of an item once it recedes beyond the n-2 horizon, as opposed to the mirror-image trace of temporarily-unattended items. Documenting that difference would add value to the study.

Although we agree with the reviewer that this is an interesting question, our current design doesn't let us address it in a satisfactory manner. First, just in terms of the status of item n-2's neural representation, we'd only be able to assess this with the two thirds of the items that are "non-matching" probes, because for "matching" probes item n-2 is the same as item n, and so this analysis would be underpowered. Furthermore, assuming that this (underpowered) analysis yielded the null reconstruction that we expect that it would, we'd have no basis for knowing whether this came about due to decay vs. an active removal mechanism, because the experimental factors that would be needed to test this hypothesis are not built into the design.

- In the Results section the authors identified two time periods during the delayed-matching task that proved "optimal" for decoding during the 2-back task, and they say that they will use the later one (940-1040 ms), but in the Hypothesis section, they speak of an earlier 440-540 ms window to be used for training the IEM. Which time window will be used?

We apologize for the confusion: the two windows the reviewer refers to are, in fact, the same, but alternatively described with reference to stimulus onset (the "940-1040 ms" window) or offset (the "440-540 ms" window). We have fixed this so that now the manuscript uses just one convention for describing time points during the task.

- Perhaps more important: Will the time window now be fixed for the preregistered study, or again chosen to be optimal based on the data from that study? The latter may or may not be problematic, depending on how exactly the criterion of optimality is defined – on which unfortunately there was very little information.

Yes, the time window (940-1040 after stimulus onset) will be fixed for the preregistered study.

- Calculating sample sizes a priori is always risky, especially when venturing into new territory. The authors did a good job in estimating effect sizes and power, but there remains the risk that the study will fall just short of a significant result on one of the main hypotheses. The authors might consider using Bayesian statistics to test the hypotheses. This would enable them to add further subjects to the sample in case the evidence remains ambiguous after N=30 (Rouder, 2014).

Rouder, J. N. (2014). Optional stopping: No problem for Bayesians. *Psychonomic Bulletin & Review*, 21, 301-308.

We agree with the reviewer in spirit, that a Bayesian approach can often be preferable to classical hypothesis testing, for a variety of reasons. In this instance, however, it would seem to us to run contrary to the rationale of preregistration if we didn't commit ahead of time to an a priori N. If we were to encounter the scenario raised by the reviewer, we suppose that our next step would be to contact the editor to consult about whether we should strictly adhere to the preregistered N, or perhaps make the post hoc decision to add additional subjects (and presumably report both the results at the a priori stopping point as well as the results from the exploratory additional testing).

Reviewer 2:

1. Is the single item delayed recognition task still necessary if you will also include a 1-back condition in the main experiment? Does the latter not allow for robust training of itself?

We agree with the reviewer's reasoning in principle. However, planning to use the 1-back data to train models would complicate the initial intention of using the 1-back data as a control task to rule out the "undershoot" alternative hypothesis. (I.e., what if the 1-back data were (unexpectedly) to show an undershoot. In this scenario we then wouldn't be able to use them to train the "early delay" model.) In addition, the delayed-recognition task cannot be dispensed with because our optimal IEM training model, as specified in the Principle Hypotheses, is derived from this task.

2. As an additional exploratory analysis I would find it interesting to see if the N-1 stimulus can be reconstructed from the non-match n stimuli, and what that reconstruction would then look like. This situation would be close to the resuscitation through unrelated stimulation scenario (even though the mask serves this function better here).

This is an interesting idea, and although we also agree that it's likely that the mask already accomplishes this, we have added this as a secondary hypothesis (#10). As the reviewer pointed out, we predict the presentation of n would elicit a transient positive reconstruction of the non-match item n-1.

3. In the pilot data, why was behavioural performance in the 2-back better than in the single item delayed recognition task? This was counterintuitive to me. Do the authors expect this to happen again in the experiment proper?

This is because the memory probe in the delayed recognition task required a finer-grained, and therefore more difficult, discrimination – nonmatching probes differed by 15° in delayed recognition vs. by a minimum of 30° in 2-back.

4. The last two p-values on p. 25 seem incorrect (I did not check all of them).

Those p values were correct; they are identical because they are from the t tests in Principal Hypotheses 1 and 2, which were FDR-corrected for multiple comparisons. We have added this point on p. 25 for clarification.

5. A study by Greene et al (2015) appears relevant as background:

Greene, C. M., Kennedy, K., & Soto, D. (2015). Dynamic states in working memory modulate guidance of visual attention: Evidence from an n-back paradigm. *Visual Cognition*, 23(5), 546-560.

We discussed Greene et al. (2015) prior to the submission of the registered report, but decided not to include this work because we find aspects of the design and results to be problematic. In their Experiment 1, they failed to include a crucial control condition in which memory items in the 2-back task were absent in the visual search, without which their hypotheses couldn't be tested decisively. Although they tried to remedy this in an Experiment 2 by including a neutral condition, the results failed to show a significant difference between the effects of 'invalid-1back' (i.e., unprioritized) and 'invalid-2back' (i.e., prioritized) items on search RT, which did not support their main hypothesis.

Reviewer 3:

1. Pilot data on which the preregistered report is based, shown in Figure 4, indicate that contrary to the default assumption that the activity representing an item in working memory might simply get weaker when it is deprioritized, the item is recoded in a different activity pattern, that may (as the authors propose) suggest that a process of priority-based recoding helps to protect remembered information when it is not in the focus of attention. Specifically, deprioritization was associated with negative/shifted IEM reconstruction. Yet, this negative/shifted orientation encoding is shown while another representation (that of n-1) is prioritized. In the new study, to more conclusively show that the negative/shifted tuning is indeed related to deprioritization/recoding of item n (rather than another orientation being represented in the activity pattern), a control analysis is necessary. One possibility is to conduct a control analysis in which the IEM is (i) trained on trial n in the 1-back task, i.e., in the absence of any deprioritization of a previously presented item, and (ii) tested on n-1 pretending it is trial n (one could also train on all even trials and test on all uneven trials with the labels of the even trials). One would expect no opposite reconstruction in this case (new hypothesis) IF the observed pattern in the 2-back task is truly related to deprioritization¹. This is important as it has alternatively been suggested that deprioritized items may not be encoded in neural activity patterns, but rather in synaptic efficiency (as the authors also discuss in the introduction).

¹ One possibility is that this control analysis of the 1-back data will actually reveal some reconstruction of the non-presented orientation at trial n-1, given that it was recently shown by Bae and Luck (in press, *Psych Science*) that an activity-silent representation of the previous trial is reactivated when the current trial begins. Yet, importantly, this should result in a positive (not negative) reconstruction.

We apologize that we are having trouble understanding this point. If we were to literally "(i) trained on trial n in the 1-back task, ... and (ii) tested on n-1 pretending it is trial n" that would be testing a hypothesis that the brain is "predicting the future," by already representing item n

at time $n-1$. We are going to assume that the reviewer mistakenly typed “ $n-1$ ” when s/he intended “ $n+1$ ”, an assumption that seems to be supported by the footnoted comment. In this case, the logic would seem to be to rule out the possibility that a negative/shifted reconstruction of an item may merely be the consequence of an item being superceded in priority by another item. That is, in the 2-back task, item $n-1$ supercedes item n as soon as the $n-2$ vs. n comparison is made; in the 1-back task, n is superceded by $n+1$ as soon as the n vs. $n+1$ comparison is made. To test this hypothesis, we would test the model of each item n (as trained on delayed-recognition data) on data from the ISI separating $n+1$ from $n+2$. Assuming that we have understood, we’ll be happy to carry out this additional analysis, and now describe it as “Secondary Hypothesis 11.” We should note, however, that this analysis suffers from the same concerns as would the analysis that we considered in response to Reviewer #1’s first point: it can only be carried out on the 2/3 of trials in which items serve as nonmatching probes (because if n and $n+1$ are matching, then it doesn’t make sense to say that n was superceded by $n+1$).

2. The authors propose to examine to what extent eye position can be used to determine the content of WM, which is important, as a previous study showed that orientation-dependent eye movements may have a systematic effect on the decoded signal (Mostert et al.). Yet, if this truly is the case also in the present study, this could provide a confound to the interpretation of the results of the EEG IEM analyses aimed at reconstructing orientation from the pattern of brain activity. As Mostert et al. pointed out: “If the eyes move, then the projection falling on the retina will also change, even when external visual stimulation remains identical. Thus, if gaze position is systematically modulated by the image that is perceived or kept in mind, then so is the visual information transmitted to the visual cortex. For example, if a vertical grating is presented and kept in VWM, then the subject may subtly move her or his gaze upward. Correspondingly, the fixation dot is now slightly below fixation, thus leading to visual cortex activity that is directly related to the retinotopic position of the fixation dot.” In other words, the IEM model may pick up on subtle differences in spatial representation (related to “attending away”), not orientation/priority-based recoding per se, potentially leading to an incorrect conclusion. Mostert et al. also propose a solution: the use of a localizer (training) task that is specifically sensitive to the neural representations encoded in bottom-up signals evoked by passively perceived gratings. Given that Wan et al.’s own pilot data also suggested that the neural code for the perceptual representation of their stimuli is the same as that for their retention in visual working memory, adding such a localizer task to the current design could provide a trained IEM that is not confounded by small eye movements/spatial representation/attending away, and allow for drawing conclusions at the level of orientation recoding in WM. Please consider adding a localizer task in which orientation is not task-relevant.

Adding such a localizer task is a good idea. However, we prefer to plan to use models built from this localizer task as supplementary to the procedures currently proposed for the primary and secondary hypothesis tests, for a few reasons. First, we prefer to keep our primary analyses as close as possible to the methods that generated the pilot results – straying too far from the initial methods risks weakening the premise that we are carrying out a replication study. Second, some recent studies on the relation to microsaccades is suggesting that they may be necessary for initiating shifts of attention and/or selection. So if it turns out, for example, that the majority of trials in the 1-back, 2-back, and delayed-recognition tasks include systematic microsaccades (despite being “clean” by conventional standards), it might be a mistake to classify them all as artifactual.*

**(For example, this recent study showed that attention-related boosts in stimulus processing was time-locked not to cues, but to the onset of stimulus-related microsaccades:*

Lowet, E., Gomes, B., Srinivasan, K., Zhou, H., Schafer, R. J., & Desimone, R. (2018). Enhanced neural processing by covert attention only during microsaccades directed toward the attended stimulus. Neuron, 99(1), 207-214.)

3. Some important details about the analyses are missing.

a) The authors do not specify if they will filter their EEG data or deal with slow drifts in the data in some other way. A study by van Driel et al. that just came out as a preprint indicates that highpass filtering can result in temporal displacement of decoding accuracy/information, and propose robust detrending as an alternative. See <https://www.biorxiv.org/content/10.1101/530220v1> Please add information about filter settings/detrending.

As in the pilot study, EEG data will be bandpass-filtered from 0.01 to 80 Hz and notch-filtered at 60 Hz. This has been added to the manuscript. As reported in the van Driel et al. paper, artifacts introduced by high-pass filtering, such as spurious decoding and temporal generalization are minimal when cut-off frequencies below 0.1 Hz are used. Therefore, our analysis plan should not be subject to these issues.

b) It is not specified if malfunctioning electrodes will be removed and reinterpolated, and if so, how.

Bad channels will be identified by visual inspection, then removed and interpolated using the 'spherical' method in EEGLAB. This information has been added to the manuscript.

c) It is not clear how trials with eye movements will be identified based on the EEG data: using the EOG traces or ICA components? And why not simply use the more accurate eye tracker data? Will trials with eye blinks during the delay period also be removed from the analysis?

Eye movements will be identified by visual inspection of the raw data as well as ICA components (see p. 18). We are favoring ICA components over eye-tracking data because this is what we did in the pilot study, which the new study is designed to replicate. Therefore, we intend to preserve as many processing and analytical decisions as possible. For the same reason, trials with eye blinks during the delay period will not be removed from analysis after blink-related artifacts are removed from the data.

d) Are only correct trials included in the training and reconstruction or are incorrect trials also included in the analyses? Please specify.

All trials (both correct and incorrect) are included in the analyses. This has been added to the manuscript.

e) It is not specified if datasets (subjects) will be excluded from the analyses based on their performance (e.g., when performing x stdev from the group average) or when too few trials are left in a given condition after cleaning. Please add criteria for exclusion if appropriate.

Subjects will be excluded from analyses if (1) their performance accuracy in any task is more than 3 standard deviations below the group average; or (2) less than 50 trials are left in any orientation condition after EEG data cleaning. This has now been added to the manuscript.

f) Sample size is now specified as n=30 based on power analyses/previous work. Does this mean data collection will continue until 30 usable datasets can be included in the analyses (i.e., will bad subjects be replaced or not; or only when less than x good datasets are left)? Please clarify.

Yes, bad subjects will be replaced and we will continue collecting data until 30 usable datasets are obtained. This has been added to the manuscript.

4. Recent work by Parthasarathy et al., suggesting code morphing in WM, not cited in the proposal, is relevant to the current ideas.

Parthasarathy, A., Herikstad, R., Bong, J. H., Medina, F. S., Libedinsky, C. & Yen, S. C. Mixed selectivity morphs population codes in prefrontal cortex. *Nature Neuroscience* 20, 1770–1779 (2017)

Parthasarathy, A et al. (bioRxiv). Time-Invariant Working Memory Representations in the Presence of Code-Morphing in the Lateral Prefrontal Cortex.
<https://www.biorxiv.org/content/10.1101/563668v1>

We now have referenced this work in our manuscript. We cannot directly bridge our preliminary findings and principal hypotheses to this work because our results are based on whole-brain EEG data from human subjects, whereas Parthasarathy and colleagues' observations come from single-neuron recordings from macaque prefrontal cortex, where neurons with different selectivity profiles display different 'code-morphing' capabilities. Nonetheless, we will keep this line of work in mind as we explore and interpret our data from the new study.

Appendix D

Reviewer 3:

1. Pilot data on which the preregistered report is based, shown in Figure 4, indicate that contrary to the default assumption that the activity representing an item in working memory might simply get weaker when it is deprioritized, the item is recoded in a different activity pattern, that may (as the authors propose) suggest that a process of priority-based recoding helps to protect remembered information when it is not in the focus of attention. Specifically, deprioritization was associated with negative/shifted IEM reconstruction. Yet, this negative/shifted orientation encoding is shown while another representation (that of $n-1$) is prioritized. In the new study, to more conclusively show that the negative/shifted tuning is indeed related to deprioritization/recoding of item n (rather than another orientation being represented in the activity pattern), a control analysis is necessary. One possibility is to conduct a control analysis in which the IEM is (i) trained on trial n in the 1-back task, i.e., in the absence of any deprioritization of a previously presented item, and (ii) tested on $n-1$ pretending it is trial n (one could also train on all even trials and test on all uneven trials with the labels of the even trials). One would expect no opposite reconstruction in this case (new hypothesis) IF the observed pattern in the 2-back task is truly related to deprioritization¹. This is important as it has alternatively been suggested that deprioritized items may not be encoded in neural activity patterns, but rather in synaptic efficiency (as the authors also discuss in the introduction).

¹ One possibility is that this control analysis of the 1-back data will actually reveal some reconstruction of the non-presented orientation at trial $n-1$, given that it was recently shown by Bae and Luck (in press, Psych Science) that an activity-silent representation of the previous trial is reactivated when the current trial begins. Yet, importantly, this should result in a positive (not negative) reconstruction.

We apologize that we are having trouble understanding this point. If we were to literally “(i) trained on trial n in the 1-back task, ... and (ii) tested on $n-1$ pretending it is trial n ” that would be testing a hypothesis that the brain is “predicting the future,” by already representing item n at time $n-1$. We are going to assume that the reviewer mistakenly typed “ $n-1$ ” when s/he intended “ $n+1$ ”, an assumption that seems to be supported by the footnoted comment. In this case, the logic would seem to be to rule out the possibility that a negative/shifted reconstruction of an item may merely be the consequence of an item being superceded in priority by another item. That is, in the 2-back task, item $n-1$ supercedes item n as soon as the $n-2$ vs. n comparison is made; in the 1-back task, n is superceded by $n+1$ as soon as the n vs. $n+1$ comparison is made. To test this hypothesis, we would test the model of each item n (as trained on delayed-recognition data) on data from the ISI separating $n+1$ from $n+2$. Assuming that we have understood, we’ll be happy to carry out this additional analysis, and now describe it as “Secondary Hypothesis 11.” We should note, however, that this analysis suffers from the same concerns as would the analysis that we considered in response to Reviewer #1’s first point: it can only be carried out on the 2/3 of trials in which items serve as nonmatching probes (because if n and $n+1$ are matching, then it doesn’t make sense to say that n was superceded by $n+1$).

REPLY REVIEWER 3: You can also run this control analysis on the $n+1$ trials. The important thing is that the control analysis demonstrates no opposite reconstruction in a condition without deprioritization.

2. The authors propose to examine to what extent eye position can be used to determine the content of WM, which is important, as a previous study showed that orientation-dependent eye movements may have a systematic effect on the decoded signal (Mostert et al.). Yet, if this truly is the case also in the present study, this could provide a confound to the interpretation of the results of the EEG IEM analyses aimed at reconstructing orientation from the pattern of brain activity. As Mostert et al. pointed out: "If the eyes move, then the projection falling on the retina will also change, even when external visual stimulation remains identical. Thus, if gaze position is systematically modulated by the image that is perceived or kept in mind, then so is the visual information transmitted to the visual cortex. For example, if a vertical grating is presented and kept in VWM, then the subject may subtly move her or his gaze upward. Correspondingly, the fixation dot is now slightly below fixation, thus leading to visual cortex activity that is directly related to the retinotopic position of the fixation dot." In other words, the IEM model may pick up on subtle differences in spatial representation (related to "attending away"), not orientation/priority-based recoding per se, potentially leading to an incorrect conclusion. Mostert et al. also propose a solution: the use of a localizer (training) task that is specifically sensitive to the neural representations encoded in bottom-up signals evoked by passively perceived gratings. Given that Wan et al.'s own pilot data also suggested that the neural code for the perceptual representation of their stimuli is the same as that for their retention in visual working memory, adding such a localizer task to the current design could provide a trained IEM that is not confounded by small eye movements/spatial representation/attending away, and allow for drawing conclusions at the level of orientation recoding in WM. Please consider adding a localizer task in which orientation is not task-relevant.

Adding such a localizer task is a good idea. However, we prefer to plan to use models built from this localizer task as supplementary to the procedures currently proposed for the primary and secondary hypothesis tests, for a few reasons. First, we prefer to keep our primary analyses as close as possible to the methods that generated the pilot results – straying too far from the initial methods risks weakening the premise that we are carrying out a replication study. Second, some recent studies on the relation to microsaccades is suggesting that they may be necessary for initiating shifts of attention and/or selection. So if it turns out, for example, that the majority of trials in the 1-back, 2-back, and delayed-recognition tasks include systematic microsaccades (despite being "clean" by conventional standards), it might be a mistake to classify them all as artifactual.*

REPLY REVIEWER 3: I understand the author's reasoning, but if microsaccades are not artifactual (which they likely are not), this renders interpretation of the neural results difficult: do changes in reconstruction reflect true changes in priority state (of an oriented grating) or (also) changes in spatial representation? A passive localizer task could help in this case. Yet, this should not stand in the way of the authors conducting the proposed study, which, as I also wrote in my original review, addresses a very topical and important question with state-of-the-art methods. I look forward to seeing its results.

Appendix E

20 May 2020

Professor Chris Chambers, Registered Reports Editor
Royal Society Open Science

Dear Chris:

We have completed the data collection and analyses for **Registered Report RSOS-190228**, as proposed in our in-principle-accepted Stage 1 submission, and are now pleased to submit the resultant Stage 2 manuscript, entitled “Tracking stimulus representation across a 2-back visual working memory task,” by Wan, Cai, Samaha, and Postle, for your consideration.

We confirm that the completed experiment has been executed and analyzed in the manner originally approved, and that only one unforeseen change was made to the approved procedures. This was a relatively minor modification of the procedure for cluster-based permutation testing to correct for multiple comparisons in timepoint-by-timepoint analyses that are effectively just descriptive, because they aren’t involved in any of the preregistered hypotheses. This change will be detailed on the page appended to this letter. Additionally, there have been several changes made to the final text of the accepted Stage 1 manuscript, and these are clearly noted using tracked changes on a copy of the original. These changes fall under three categories – grammar, style, and formatting; typographical errors; and additional methodological detail added for clarity – and these are also summarized on the page appended to this letter, with page numbers associated with each specific change noted.

The URL for raw and processed data, and for analysis scripts on the Open Science Framework can be found on page 21 of this Stage 2 manuscript. The URL for the approved Stage 1 protocol on the Open Science Framework can also be found on page 21 of this Stage 2 manuscript.

We confirm that for the primary Registered Report, no data for any pre-registered study other than pilot data included at Stage 1 was collected prior to the date of IPA. For the secondary Registered Report of existing data, we confirm that no data other than pilot data included at Stage 1 was subjected to the pre-registered analyses prior to IPA.

With thanks in advance for your consideration of this submission, and on behalf of my coauthors,

Brad Postle

Summary of changes between accepted Stage 1 manuscript and Stage 2 manuscript

Cluster-based permutation testing

- Time-point-by-timepoint significance testing, although not part of any hypothesis test, was carried out to illustrate the time course of representational transformations. The procedure used for cluster-based permutation of the timepoint-by-timepoint significance testing of the pilot data, illustrated as part of Figure 3 and Figure 4 of the accepted Stage 1 manuscript, followed an erroneous procedure of using “the largest cluster (i.e., with the most timepoints)” to construct the null distribution of the cluster-level statistic. The correct procedure was used for the Stage 2 analyses and is described in the manuscript as using “the largest cluster-level statistic (in absolute value)” to construct the null distribution (see page 26-27). We have also applied this correct procedure to the pilot data, and the legends to the updated figures (now in Supplementary Online Materials) note that this changed procedure resulted in no change to the cross-validation results for the delayed-recognition task (Supplementary Figure 1) and the loss of significance of one small epoch from each of the ISIs of the 2-back task (3180-3220 ms from item n onset and 1330-1370 from item $n + 1$ onset; Supplemental Figure 2).

Grammar, style, and formatting

- Verb tenses have been updated where appropriate;
- References to figures that have been moved have been relabeled;
- All instances of “msec” have been changed to “ms,” for consistency;
- Whereas the Stage 1 manuscript contained methods for both the pilot study that was the basis for this Registered Report and for the Registered Report itself, the Stage 2 manuscript only presents the methods for the Registered Report. Therefore, the tracked changes in the Methods section show many changes, including large blocks of text that have been removed and/or inserted. These are either instances of text describing the pilot study, which was truly deleted, or instances where text has been moved from one part of the Methods section to another. Moving text sometimes also necessitated some reformatting and minor adjustments of verbiage, for clarity.

Typographical errors – *flagged explicitly in the Stage 2 manuscript with footnotes*

- Accepted Stage 1 submission mistakenly stated, as part of *Secondary Hypothesis 9*, that the delay period of the delay-recognition task spanned from 1150-3150 ms after the target item’s onset. These are the correct values for the 1-back task, and they were presumably mistakenly copied and pasted from *Secondary Hypothesis 7*, which relates to the 1-back task. The correct values are 1000-2000 ms (see page 13).
- Accepted Stage 1 submission mistakenly stated that the radius of the stimuli was 5°. The correct dimension is 2.8° (see page 6).
- Not technically a typo, but citations of “Yu and Postle (unpublished)” have been updated to “Yu, Teng and Postle (in press).”

Additional methodological detail *While preparing the Stage 2 manuscript it came to our attention that portions of the Methods section of the accepted Stage 1 manuscript did not include sufficient detail. Therefore we have added detail pertaining to the following:*

- Blocks of the functional localizer task were interleaved with the blocks of 1-back and 2-back (see page 17).
- Each 1-back block had 127 stimuli (see page 19).
- The eight blocks of the delayed-recognition task each had a different unique randomized sequence of 72 trials (see page 20).
- Specification of the cutting of data (including baseline) into discrete “trials” for 1-back and functional localizer tasks (see page 21).
- Specifying that “All epochs were baseline-corrected and re-referenced to the median” (see page 21).
- IR-based eye-tracking data: Additional detail about preprocessing and procedures for classification analysis and significance testing (see pages 23 and 29).

Appendix F

Response to Comments on RSOS-190228.R2: Original text is re-presented here, verbatim, in this serifed font, and the authors' replies are interleaved, where appropriate, *in this italicized sans-serif font*.

Comments to Author:

Reviewer: 1

Comments to the Author(s)

This is an excellent manuscript. The authors carefully carried out the registered experiment, made their hypotheses explicit, and reported the relevant results in a transparent manner. The discussion is reasonable and the conclusion convincing. I have only a few comments on the writing that I hope will help the authors to improve the clarity of this report:

(1) When presenting their hypotheses, the authors repeat after each hypothesis "The precise method we will use..." – it is enough to say this one (for all hypotheses). It is tiring to read this over and over again.

Done.

(2) Secondary Hypothesis 10 is described in the text as reconstruction of item $n-1$ during presentation of non-matching item n . In Figure 7a, the relevant results are shown as reconstruction of item n during presentation of non-matching item $n+1$. This is the same, of course, but the mismatch in labeling is confusing.

We have changed this description of Secondary Hypothesis 10 to state "The IEM reconstruction of item n 's orientation from the EEG signal from the 2-back task during the presentation of the non-match item $n + 1$ will be ...", in order to be consistent with Figure 7A.

Perhaps more important: How do the data presented in Figure 7a differ from the data in the second period (after onset of stimulus $n+1$) in Figure 4? And related, how does Principal H 2 differ from Secondary H 10?

Figure 7A differs from the second half of the heat map in Figure 4 in that Figure 7A only includes epochs where $n + 1$ is a nonmatch to n , because the corresponding Secondary Hypothesis 10 tests whether even "unrelated" visual stimulation (i.e. the presentation of item $n + 1$) could "recode a UMI back into its 'perceptual representational format,'" whereas Primary Hypothesis 2 (Figure 4) aims to test whether stimulus n could be positively reconstructed during the delay

following the presentation of $n + 1$, so all epochs were used regardless of whether n matches $n + 1$.

(3) On p. 22, should the power of 6 not be added after “(x)”?

No. Actually powers of trigonometric functions should be placed right after the operators, e.g., ‘sin’ or ‘cos’.

(4) On p. 27, the first equation needs a closing parenthesis after x .

Fixed.

Reviewer: 2

Comments to the Author(s)

Review of RSOS-190228.R2 Wan, Cai, Samaha, & Postle, “Tracking stimulus representation across a 2-back visual working memory task”, as submitted to Royal Society Open Science

I reviewed the preregistered report for this study, and I think it has worked out really nicely. Not all hypotheses have been confirmed, but that would have been a stretch, and the exact reason why a preregistration is so nice. The essential predictions were confirmed, making this study very relevant. I recommend publication, and I congratulate the authors with a very nice study.

I do have a few remarks/questions. I do not need to see the paper again, but the authors may take these along in a final version.

In the Discussion (see also Hypothesis 1) it is argued that the unprioritized memory (UMI) is held in an active representation. By that I assume a patterns of neural firing activity. This a) is a bit odd, since later on in the discussion it is argued that the prioritized memory (PMI) is represented silently; and b) strikes me as unnecessary, as the UMI could still be encoded in connectivity pattern, and what is being picked up is random or nonspecific activity flowing through the network? In any case, unless I’m missing the logic, I think the authors would do well to elaborate on why they assume the UMI to be active, and the PMI passive...

We believe that this point is inconsistent with the unambiguous construal of what “active” and “activity” are understood to mean in the Introduction, and that have been so since the review of the Stage 1 manuscript. In the Introduction the explicit understanding is that an “active representation” is inferred when it can be

reconstructed from the EEG signal, and a “silent” representation is inferred when such reconstruction is not successful but logic dictates that the item is in working memory (as has been the case, in previous studies, for the UMI). To make this clearer than it already is (and presumably already has been since Stage 1), we’d need to add verbiage like this to the Introduction. This would violate the journal’s policy of leaving the Stage 1-accepted Intro. untouched with the exception of minor tweaks to syntax, such as changing verb tense. It follows from this that the reviewer’s point “b)” is not a valid proposition. Furthermore, point “a)” here must also reflect a misunderstanding on the reviewer’s part, because it is simply not correct: the Discussion never states that the PMI is represented solely in a silent format (which is what we understand this statement to be implying.) Rather, it is very clear throughout the manuscript that the PMI is assumed to always be represented in an active state. The additional nuance is the possibility that the PMI may also be represented, in parallel, in silent code. This idea is not ours, but rather has been simulated explicitly in two computational studies that we cite (Manohar et al. and Masse et al., both 2019).

On p. 38 it is argued that instead of re-coding, we better think of it as re-mapping. However, I found it difficult to follow the argument here. That may well be me being thick, but if the authors see a way of clarifying their argument further that would be great.

We have re-read the relevant passage several times, and frankly aren’t sure how this can be made clearer. Because this is an important conceptual “take-home” from this paper, however, we’ve decided to err on the side of potential redundancy by adding this sentence to this section on p. 38: “Stated another way, when a stimulus transitions to an unprioritized status, the set of mappings between stimulus values and neural patterns rotates such that the individual mappings are now different, but the distance (in orientation) between neural patterns, and therefore the neural code, is preserved (Yu, Teng, & Postle, 2020).”

Minor things:

p.4, bottom: “reconstructed from signal from early visual cortex”

This is a grammatically correct clause that communicates precisely what we intend to convey, and so we have not changed it.

p.6, middle “quality of the task that make it well-suited”

Fixed.

p.23 the conversion of eye x,y coordinates to “visual angle” is not clear. To visual angle as in distance, or to polar coordinates? And if the latter, then only angle or also eccentricity? And if polar angle only, how can we have outliers then? And if it’s been converted to simply visual angle (as in distance), it’s unclear why that adds anything to the analyses.

The converted visual angles indicate “distances” from fixation along the x and y axes, not polar coordinates. In the current study, as the reviewer pointed out, using x and y coordinates measured in number of pixels is indeed equivalent to using visual angle and “doesn’t add anything to the analyses,” but that is only because we used the same monitor with the same resolution for all sessions and the distance from the screen was kept constant across subjects. If either of these factors were not the same, raw x and y coordinates in number of pixels wouldn’t be a consistent measure across subjects, which might pose a problem for classification analyses; however, this problem would be avoided by using visual angle, which would still be a consistent measure. In addition to this concern, visual angle is a more sensible measure for eye movement (i.e., the angle of rotation of the eyes) than number of pixels on the monitor screen.

Signed,

Chris Olivers

Reviewer: 3

Comments to the Author(s)

This is a well written and carefully reasoned paper, providing results from a preregistered EEG study addressing a key outstanding question: How does the neural representation of visual working memory content vary with behavioral priority? The finding that items in working memory that are currently unattended are encoded in an opposite/negative representational format is important as it suggests that priority-based remapping helps to protect this remembered information when it is not yet relevant to the task at hand.

I can confirm that the introduction, rationale and stated hypotheses are the same as the approved Stage1 submission and that the authors adhered precisely to the registered experimental procedures and analyses. The exploratory functional-localizer based analyses are justified, sound, and informative. The authors’ conclusions are justified given the data. The observation that an unprioritized item in working memory can be encoded in a distinctive “opposite” representational

format compared to a prioritized item provides an important contribution to the rapidly growing literature on the neural mechanisms underlying prioritization of information in working memory.

I only have several more minor comments that the authors might wish to consider.

1. The main finding is that the UMI is associated with negative reconstruction of orientation (Fig. 4), and that this cannot be explained by a post-stimulus undershoot, as this effect is not observed in the 1-back task (Fig. 5a). I did not think of this during the Stage1 review, but it seems critical to me to directly statistically contrast these two effects. Is the reconstruction of the UMI orientation in the 2-back task statistically different from that in the same period in the 1-back task? Although an exploratory analysis at this point, this could strengthen this theoretically interesting finding.

As the reviewer points out, our inclusion of the 1-back task aimed to rule out the post-stimulus undershoot account for the negative reconstruction of UMI in the 1-back task. However, we do not think directly statistically contrasting the two effects would be a stronger test in place of the one proposed in Principal Hypothesis 3. Even if the reconstruction of stimulus n in the ensuing delay turned out to be significantly different between the 2-back and 1-back tasks, it would still be possible for stimulus n to be negatively reconstructed in 1-back, similar to 2-back but with a different magnitude, which would still be consistent with the undershoot account. In contrast, the failure to reconstruct stimulus n in 1-back from a direct test, as detailed in the paper, provides stronger evidence that the negative reconstruction of UMI in 2-back cannot be accounted for by a post-stimulus undershoot.

2. A surprising aspect of the current findings is the lack of robust decoding of the PMI in the 2-back task. I think it is important to discuss potential explanations for this unexpected observation, even if speculative.

One possibility is that an item is reprioritized in a different format/neural code (e.g., a verbal vs. visual code). An often-used strategy in n-back tasks in particular at higher levels of n is to verbally rehearse the items to be kept in WM. Although orientation is less easily verbalized than for example letter identity, given that only 6 widely-spread orientations were used, it is possible to verbally label them (e.g., they could be seen as pointing to a position on a clock (e.g., 2 o'clock -> "two")). Could one explanation for some of the current findings then be that (some) participants (at least on some portion of trials) recoded the orientation of the

memory items into a verbal format during the 2-back task? I don't readily see how verbal recoding could have led to the observed negative reconstruction of the UIM, but item n may have been recoded to a verbal format after deprioritization to prevent interference from new visual input. This could have subsequently reduced the ability to reconstruct the item when reprioritized in the 2-back task, and could possibly also explain the lack of transfer from the functional localizer trained model to the 2-back task.

The first explanation is a possible account of the failure to reconstruct the PMI, although we did try to discourage verbalization in two ways: (1) during the 2-back task training session and at the beginning of the EEG session, subjects were explicitly instructed to not to resort to verbal labeling to remember the stimuli; (2) we intentionally avoided cardinal orientations for the grating stimuli to discourage verbal strategies. The further speculation about the UMI remapping effect, however, is not consistent with this line of reasoning, because it is with the delayed recognition-trained model – capturing the code that, by this logic, the PMI may have been recoded “away from” – that the UMI reconstructs in an opposite manner.

Another possibility is that the PMI is more “action-oriented” in nature than (some of) the representations the EIM models were trained on, and also includes contributions from more frontal regions (Myers et al., TiCS, 2017).

Although we can't rule this possibility “in” or “out” with any certainty, at a literal level of interpretation it strikes us as unlikely because the delayed-recognition task on which the IEMs were trained had the same action/motor demands as did the 2-back task: pressing a key to indicate match/non-match between the sample and the probe. At a more general level, however, it is certainly the case that at least some contextual factors differed between the two tasks – one obvious one is that they were performed at different times. However, these differences also apply to the representation of the UMI during the 2-back task, and this then requires speculation about why some contextual factors may have been more important for the PMI than the UMI, and we don't feel that it would be useful to engage in such wide-ranging speculation in this manuscript that otherwise addresses very concrete hypotheses.

A final possibility is that the number of neurons coding the reprioritized PMI is much smaller than the original population used for initial encoding/representation, and hence that there was an active PMI trace, but it was beyond the detection threshold of non-invasive techniques such as EEG. Invasive recordings in humans support this possibility (Kornblith et al., Current Biology, 2017).

This is a possibility that we can neither rule in nor out, but if it were true, it would raise a new question, which is why it presumably wasn't a factor in the Pilot Study, which featured fewer subjects but nonetheless produced significant reconstructions of the PMI. Rather than entertain an untestable possibility that would then necessitate even more speculation about its accompanying "new question," we have chosen to add an additional sentence that points to an objective fact: "Of possible relevance is the fact that one difference between the Pilot Study, in which the PMI was positively reconstructed during the 2-back task, and the Registered Report, is that in the former 2-back task blocks were tested consecutively, whereas in the latter they were interleaved with other tasks, a factor that may have added noise, thereby decreasing sensitivity of the analysis."

3. There is also monkey work showing that some neurons in inferior temporal cortex and V1 remain persistently active during VSTM maintenance (Supèr & Ran, 2008; Supèr et al., 2001; Woloszyn & Sheinberg, JoN, 2009).

Yes, of course, but we prefer not to stray into a wide-ranging review of the neurophysiology of visual working memory, which could (and has) filled many lengthy review papers.

4. A study by Olivers and colleagues (De Vries, I. et al., Neuroimage, 2019) previously used MVPA to decode memory status (current vs. prospective) from the pattern of scalp EEG activity during a working memory task. This study and its findings seem relevant.

Thanks for the reference. The De Vries et al. (2019) study indeed included the manipulation of priority. However, their main interest is the top-down control signal involved in prioritization (decoding priority status) rather than the representations of differentially prioritized stimuli (decoding stimulus identity), which is the focus of this paper. So we prefer not to include this study but will take it into account in future study designs.

5. This is more of a side thought that may be important for future studies: a grating at fixation covers four quadrants of the visual hemifield, which given the anatomy of V1 (calcarine sulcus) could lead to a very diffuse projection of V1 activity at the level of the scalp. Not ideal for detecting weak effects. (see e.g., Fig 1 in Vagenas, MI et al., Journal of Neural Engineering, 2013)

Thanks for the interesting suggestion.

6. For completeness, Figure 8 should also show the results of the FL-trained IEM results for the 2-back task data.

Because this relates to an exploratory analysis added after the Stage 1 manuscript was accepted, we think it more appropriate to keep these in Supplementary Materials. Of course, at the editor's discretion, we could update Figure 8 along these lines.

7. Page 5: The abbreviation AMI is used twice. This should be PMI.

Fixed, thank you.

8. The color coding used for the figures 3-5, 7-8 is somewhat confusing. Blue colors can indicate both positive and negative reconstruction. Moreover, which color denotes 0 and hence which colors signal positive or negative correlations, changes from figure to figure (e.g., in Figure 3, 0 = blue; in Figure 4, 0 is more greenish-blue; and in Figure 5b, 0 is dark blue). Please make the color coding consistent across figures and so that e.g., green/blue = negative values and yellow/red = positive values.

We regret the confusion, but are reluctant to make the implied changes, because to do so would be to sacrifice the clarity of individual figures in the interest of imposing a consistent scheme across all figures. Because channel responses are measured in arbitrary units, what matters for each IEM reconstruction heat map is the relative warmth of the colors rather than the specific color values. The current color scales are automatically generated by a MATLAB function to optimize the ability to discriminate among the full range of channel responses in each figure.